# LEARNING PERFORMANCE-IMPROVING CODE EDITS

**Alexander Shypula**[1]*, **Aman Madaan**[2]*, **Yimeng Zeng**[1], **Uri Alon**[4],
**Jacob Gardner**[1], **Milad Hashemi**[3], **Graham Neubig**[2], **Parthasarathy Ranganathan**[3],
**Osbert Bastani**[1], **Amir Yazdanbakhsh**[4]

[1]University of Pennsylvania, [2]Carnegie Mellon University, [3]Google [4]Google DeepMind
shypula@seas.upenn.edu and ayazdan@google.com

## ABSTRACT

With the waning of Moore's law, optimizing program performance has become a major focus of software research. However, high-level optimizations such as API and algorithm changes remain elusive due to the difficulty of understanding the semantics of code. Simultaneously, pretrained large language models (LLMs) have demonstrated strong capabilities at solving a wide range of programming tasks. To that end, we introduce a framework for adapting LLMs to high-level program optimization. First, we curate a dataset of performance-improving edits made by human programmers of over 77 K competitive C++ programming submission pairs, accompanied by extensive unit tests. A major challenge is the significant variability of measuring performance on commodity hardware, which can lead to spurious "improvements". To isolate and reliably evaluate the impact of program optimizations, we design an environment based on the gem5 full system simulator, the de facto simulator used in academia and industry. Next, we propose a broad range of adaptation strategies for code optimization; for prompting, these include retrieval-based few-shot prompting and chain-of-thought, and for finetuning, these include performance-conditioned generation and synthetic data augmentation based on self-play. A combination of these techniques achieves a mean speedup of $6.86\times$ with eight generations, higher than average optimizations from individual programmers ($3.66\times$). Using our model's fastest generations, we set a new upper limit on the fastest speedup possible for our dataset at $9.64\times$ compared to using the fastest human submissions available ($9.56\times$).[1]

## 1 INTRODUCTION

Despite the impressive progress of optimizing compilers and other tools for performance engineering (Aho et al., 2007), programmers are still largely responsible for high-level performance considerations such as algorithms and API choices. Recent work has demonstrated the promise of deep learning for automating performance optimization (Garg et al., 2022; Mankowitz et al., 2023). However, these techniques are either narrow or difficult to build on due to the lack of open datasets and lack of reliable performance measurement techniques, which has stymied research in this direction. Recently, pre-trained large language models (LLMs) have demonstrated impressive performance at a wide range of programming tasks (Chen et al., 2021b; Fried et al., 2022; Xu et al., 2022; Nijkamp et al., 2022). Yet, the effectiveness of large, pre-trained LLMs for program optimization remains an open research question. We study whether such LLMs can be adapted for performance optimization. To this end, we introduce a novel benchmark for performance opti-

---

*Equal contribution.
[1]Code: https://pie4perf.com

Figure 1: An example of a program that solves the problem of "*compute the sum of the numbers from 1 to N*". The program on the left runs in $\mathcal{O}(N)$, whereas the program on the right runs in constant time. The goal of PIE is to enable LLMs to perform these kinds of program optimizations.

mization that addresses the key challenge of replicable performance measurement, and perform an extensive evaluation of a wide range of adaptation techniques based on it.

First, we construct a dataset of **P**erformance-**I**mproving **E**dits (PIE). We collect C++ programs written to solve competitive programming problems, where we track a single programmer's submissions as they evolve over time, filtering for sequences of edits that correspond to performance improvements.

Next, a major challenge is the significant variability of measuring performance on real hardware due to server workload and configuration issues. Indeed, we find that benchmarking on real hardware can lead to large, phantom performance "improvements" due only to random chance. To address this challenge, we evaluate performance using the gem5 CPU simulator (Binkert et al., 2011), the gold standard CPU simulator in academia and industry, and models state-of-the-art general-purpose processors. This evaluation strategy is entirely deterministic, ensuring both reliability and reproducibility.

Based on this benchmark, we evaluate a variety of techniques for adapting pre-trained code LLMs for performance optimization. First, we consider baseline prompting approaches, including techniques such as chain-of-thought (Wei et al., 2022b) (CoT). We find that LLMs are limited in the challenging task of code optimization. Without data-driven methods that leverage PIE, our strongest baseline CoT only warrants a $1.60\times$ average speedup over 8 submissions vs. the $3.66\times$ human reference. Next we consider a retrieval-based prompting approach where retrieval is used to select examples most similar to the current one (Liu et al., 2021; Poesia et al., 2021). Lastly, we consider several finetuning strategies: these include using synthetic data generated via self-play (Haluptzok et al., 2022), where synthetic training examples are generated by an LLM without the need for direct human examples, as well as performance-conditioned generation, where we condition generation on the performance of the generated program.

We find that data-driven methods using PIE, like retrieval-based few-shot prompting and fine-tuning, are highly effective at achieving strong optimization abilities in LLMs. When allowing a model to take 8 samples and filtering for correctness and execution time, our fine-tuned performance-conditioned version of CODELLAMA 13B can achieve an average speedup of $5.65\times$ on our test set, and a fine-tuned version of GPT-3.5 augmented with synthetic data via self-play achieves an average speedup of $6.86\times$, the average individual human sampled in our test set achieved an average speedup of $3.66\times$. Aggregating over all human submissions in the test set with a higher sampling budget, GPT-3.5 achieved a speedup of $9.64\times$ and surpassed the best human submissions across all programmers with a speedup of $9.56\times$. In summary, our contributions are:

- We introduce a new code dataset of more than 77 K C++ program pairs, named PIE, with execution time annotations collected from the gem5 simulator. PIE enables reproducible evaluation of LLMs for program optimization and reliable performance annotations for training.

- Enabled by our benchmark, we evaluate different prompting and fine-tuning approaches for adapting pre-trained LLMs to optimize programs. Our results indicate that pre-trained code LLMs are limited in their ability to optimize code without a dataset like PIE.

- We develop three effective strategies for adapting LLMs for code optimization: retrieval-based prompting, performance-conditioning, and self-play. Overall, our best model, GPT-3.5 augmented with synthetic data obtained from self-play, achieves an average speedup of 6.86×, and optimizes 87.63% of the test set by at least 10%.

## 2 PERFORMANCE IMPROVING EDITS (PIE) DATASET

We construct a dataset targeted at adapting code LLMs to performance optimization, focusing on optimizing program execution time. Our dataset is constructed based on performance-improving edits (PIE) made by human programmers in a range of competitive programming tasks from CodeNet (Puri et al., 2021). We exclusively focus on C++ programs since it is a performance-oriented language compatible with the gem5 simulator. Given a problem, programmers typically write an initial solution and iteratively improve it. Let $\mathbb{Y}^u = [y_1^u, y_2^u, ...]$ be a chronologically sorted series of programs, written by user $u$ for problem $x$. From $\mathbb{Y}^u$, we remove programs that were not accepted by the automated system, eliminating incorrect programs (fail one or more unit tests) or take more than the allowed time to run, resulting in a *trajectory* of programs $\mathbb{Y}^* = [y_1^*, y_2^*, \ldots, y_n^*]$.

For each trajectory $\mathbb{Y}^*$, we construct pairs $\mathbb{P} = (y_1, y_2), (y_1, y_3), (y_2, y_3) \ldots$, and keep only pairs for which $\frac{(\texttt{time}(y_i) - \texttt{time}(y_{>i}))}{\texttt{time}(y_i)} > 10\%$ where $\texttt{time}(y)$ is the measured latency of program $y$ (i.e., the relative time improvement is more than 10%). The CodeNet dataset includes CPU time, but we found the information to be inconsistent (see Appendix A.9). Thus, we relabel the execution time using gem5 as described below; to create these annotated runtimes, we performed over 42.8 million simulations in our gem5 environment.

We split the resulting dataset of pairs $\mathbb{P}$ into train/validation/test sets, ensuring that any particular competitive programming problem only appears in one of them. We obtain a training set of 77,967 pairs from 1,474 problems, a validation set of 2,544 pairs from 77 problems, and a test set of 978 pairs from 41 problems. For each pair in the test set, we also record the fastest human submission execution time for that problem; in Section 4, we include this running time as a comparison point.

**Test cases.** Our goal is to improve performance while ensuring correctness. We evaluate correctness through unit tests; we reject the program if a single test fails. CodeNet includes an average of 4 test cases per problem. To improve coverage, we include additional test cases from AlphaCode (Li et al., 2022) generated with a fine-tuned LLM. A small set of test cases would lead to substantial timeouts above 2 minutes in gem5; after excluding them, we obtain a median of 82.5 test cases per problem in our training set, 75 test cases per problem in our validation set, and 104 test cases per problem for our test set (Appendix A.5).

**Performance measurement using gem5.** Benchmarking program performance is notoriously difficult. For instance, code instrumentation introduces overhead, and there is substantial variance across executions due to factors such as server load and idiosyncrasies introduced by the operating system. If benchmarking is not performed carefully, it is easy to mistakenly over-report program optimization results. With enough samples and variance, benchmarking the same exact program can easily lead us to report significant optimizations.

To illustrate the challenges, consider HYPERFINE (Peter, 2023), a Rust library designed to precisely benchmark binaries. We benchmarked 500 programs "pairs" where the "slow" and "fast" programs are identical. Ideally, we should have $\frac{\text{source time}}{\text{target time}} = 1$ (i.e., the two programs have identical performance). However, we

observed a mean speedup of $1.12\times$, with a standard deviation of 0.36, and the top 5% of pairs exhibited a speedup of $1.91\times$. These results underscore the significant challenges in performance measurement.

To address this challenge, we measure program performance using the gem5 (Binkert et al., 2011) full system simulator, which provides detailed microarchitectural emulation of modern processors. Executing deterministic programs in gem5 provides deterministic performance results. This enables reproducibility in research and denoised performance measurements.[2] We use the `Verbatim` configuration of the Intel Skylake architecture from gem5.[3] An advantage of this approach is that our framework can be applied to other platforms like ARM or RISC-V without having access to hardware for those platforms.

## 3 ADAPTING CODE LLMs TO PROGRAM OPTIMIZATION

### 3.1 FEW-SHOT PROMPTING

**Instruction-prompting.** We use prompts instructing the LLM to improve the performance of the given program, an approach commonly referred to as instruction prompting (Mishra et al., 2021; Gupta et al., 2022; Longpre et al., 2023); details on the prompt are in Figure 12 in Appendix A.11.

**Few-shot prompting.** Next, we use few-shot prompting (Brown et al., 2020). In particular, we create a prompt with the format "slow$_1 \rightarrow$ fast$_1 \parallel$ slow$_2 \rightarrow$ fast$_2 \parallel \dots$". A slow test set program is appended to this prompt during inference and supplied to the model. We create the prompts by randomly sampling two (fast, slow) pairs from the training set (Examples of prompts in Figure 13 in Appendix A.11).

**Chain-of-thought prompting.** Inspired by CoT (Wei et al., 2022b), we designed prompts that ask the LLM to *think about* how to optimize the program before actually producing the optimized program. This strategy is used in conjunction with few-shot prompting (Examples of prompts in Figure 14 in Appendix A.11).

**Dynamic retrieval-based few-shot prompting.** Recent work has demonstrated that retrieval-based mechanisms can improve language models for various tasks requiring factual or procedural knowledge (Liu et al., 2021; Poesia et al., 2021; Rubin et al., 2022; Madaan et al., 2022; Shrivastava et al., 2023). Program optimization is a non-trivial task requiring knowledge of algorithms, data structures, and programming grounded in performance; thus, retrieving highly relevant examples may improve an LLM's optimization ability. For example, a solution optimized for a knapsack problem in dynamic programming could inform strategies for the coin change problem. Through dynamic retrieval-based prompts, we aim to match tasks with analogous structures or challenges, allowing models to better harness the patterns in PIE. We use the CodeBertScore models trained for $C++$ (Zhou et al., 2023b) to embed both the program to be optimized and the programs in PIE. We use FAISS (Johnson et al., 2019) to retrieve $K$ closest programs from the training set; and to construct a "slow$_1 \rightarrow$ fast$_1 \parallel$ ..." style prompt on the fly (See Figure 15 in Appendix A.11).

### 3.2 FINETUNING

We also consider fine-tuning to improve pretrained code LLMs using our PIE dataset. In addition to standard fine-tuning on the entire dataset, we describe additional strategies we used.

**Dataset imbalance.** While we have tens of thousands of slow-fast pairs in the PIE training dataset, these submissions target just 1,474 problems, which may limit the learned model's ability to generalize to new programs. Furthermore, submissions are not uniformly distributed across problems. To address this imbalance, we additionally fine-tune with a subset of 4,085 "high-quality" slow-fast pairs—in particular, we take

---

[2]This assumes gem5 terminates. Our experiments use a two-minute timeout, which may introduce slight variability. Note that altering this timeout could change the results.

[3]`https://github.com/darchr/gem5-skylake-config`

examples with the highest speedup and disallow more than 4 submission pairs per problem, for an average of 2.77 pairs per problem. Given the high costs of training models through the OpenAI API, we also use this dataset as a base for fine-tuning experiments with GPT-3.5.

**Performance-conditioned generation.** Programs can typically be written in many ways with different performance profiles. Consequently, when training a model to predict performance-improving edits with a large dataset like PIE, it is trained on a mix of large and small improvements, without any information on which improvements are more desirable than others. Inspired by recent prompting strategies (Zhang et al., 2023) and offline-rl (Chen et al., 2021a), we introduce performance tags during training by associating each "fast" program with a tag indicating the optimal achievable performance across all solutions in the dataset. Specifically, the tag indicates how close that program is to peak performance on a binned-scale $\{1, 2, \ldots, 10\}$. We instantiate our tags by categorizing the top 10% of optimized solutions in the dataset for a given task as "10/10", the next 10% as "9/10", and so on. These tags enable the model to discern the relationship between specific problem attributes and their corresponding high-performance solutions (Figure 2, left). During inference, we prompt the model with a test input and a maximal score tag "10/10", directing it to generate a highly-optimized solution (Figure 2, right).

```
This is a slow program we want to
↪  optimize to score
↪  {score_tag}/10.

### Program:
{src_code}

### Optimized Version with score
↪  {score_tag}/10:

### Optimized Version:
{fast_code}
```

```
This is a slow program we want
↪  to optimize to score 10/10.

### Program:
{src_code}

### Optimized Version:
```

Figure 2: Training (left) and inference (right) prompts for Goal-Conditioned optimization with PIE.

**Synthetic data.** Given the high cost of obtaining human-written programs, we also augment our dataset with synthetic examples through a multi-stage process. First, we prompt OpenAI's GPT-3.5 with examples from the PIE dataset, instructing it to produce novel competitive programming problems. After filtering out programs with input/output behavior identical to those in PIE and tracking semantic duplicates among those generated, we obtain 3,314 unique synthetic programs and many thousand more duplicates. Using these novel programs, we then generate an optimized version for each synthetic program using a GPT-3.5 model that has been fine-tuned on the original PIE dataset. Finally, we keep pairs where the optimized program is at least $5\times$ faster and limit semantic duplicates to three, resulting in 1,485 optimized synthetic examples. This methodology aligns with self-play and self-instruct approachs in neural program synthesis (Haluptzok et al., 2022; Rozière et al., 2023). We provide additional details on the generation process in Appendix A.6.

## 4 EXPERIMENTS

**Models.** We evaluate and adapt models from the CODELLAMA models (Rozière et al., 2023) and models from OpenAI available through their API. We also used pretrained checkpoints of (Rozière et al., 2023): CODELLAMA {7B,13B,34B} obtained via HuggingFace (Wolf et al., 2020). For the CODELLAMA family of models, we use the base set of models that have not been instruction-tuned, as the authors of the paper note that instruction-tuning diminished the performance on code generation. We provide training details

Table 1: **Summary of Results:** This table reports our strongest performing models by SPEEDUP across different adaptation regimes covered in subsequent sections. We report results for Open-access CODELLAMA **(O)** and black-box or private OpenAI models **(P)**. The highest number in each column is **bolded** and the second-highest is underscored.

| Scenario | Model | %Opt | Speedup | %Correct |
|---|---|---|---|---|
| Human reference | | 100.00% | 3.66× | 100.00% |
| Prompt **(O)** | CODELLAMA 34B, CoT | 19.63% | 1.30× | 78.73% |
| Prompt **(P)** | GPT-3.5, CoT | 43.05% | 1.60× | 91.72% |
| Retrieval **(O)** | CODELLAMA 34B | 42.54% | 2.43× | 73.62% |
| Retrieval **(P)** | GPT4 | 76.07% | 3.93× | **95.71%** |
| FineTune **(O)** | CODELLAMA 13B-PC | 66.56% | 5.65× | 70.96% |
| FineTune **(P)** | GPT-3.5, SP | **87.63%** | **6.86×** | 95.09% |

in Appendix A.8. We experiment with `gpt-3.5-turbo-0613` by prompting the pre-trained model and using the fine-tuning API. We evaluate `gpt-4-0613` by prompting the pre-trained model. During the drafting of this paper, fine-tuning GPT4 was not available through the API.

**Metrics.** To evaluate performance, we measure the following for functionally correct programs:

- **Percent Optimized** [%OPT]: The fraction of programs in the test set (out of 978 unseen samples) improved by a certain method. A program must be at least 10% faster and correct to contribute.
- **Speedup** [SPEEDUP]: the absolute improvement in running time. If $o$ and $n$ are the "old" and "new" running times, then $\text{SPEEDUP}(\text{O}, \text{N}) = \left(\frac{o}{n}\right)$. A program must be correct to contribute.
- **Percent Correct** [%$Correct$]: The proportion of programs in the test set that are at least functionally equivalent to the original program (included as a secondary outcome).

As described in Section 2, we count a program as functionally correct if it passes every test case in our dataset. Though correctness is not our primary focus, we include it to help interpret our results. In addition, we report our SPEEDUP as the average speedup across all test set examples. For generations that are either incorrect or slower than the original program, we use a speedup of 1.0 for that example, given that, in the worst case, the original program has a speedup of 1.0. We benchmark performance using our gem5 environment and all test cases mentioned in Section 2. We compile all C++ programs with `GCC` version 9.3.0 and C++17 as well as the `-O3` optimization flag; therefore, any reported improvements would be those on top of the optimizing compiler.

**Decoding strategy.** Code generation is known to benefit from sampling multiple candidate outputs for each input and choosing the best one (Li et al., 2022); in our case, "best" is the fastest program that passes all test cases. We use BEST@$k$ to denote this strategy with $k$ samples and a temperature of 0.7. We present an overview of our results in Table 1, our baseline results in Table 2, and retrieval-based few-shot prompting results in Table 3. We also calculated the upper limit for speedup for the test set by aggregating the fastest submission across all submissions from CodeNet (118,841 accepted solutions) versus the fastest generation from our fastest model when increasing samples to 40 in Section 3.2.

## 4.1 RESULTS FOR FEW-SHOT PROMPTING

**Baseline few-shot prompting.** Table 2 shows results on few-shot prompting techniques (Section 3.1, prompts in appendix A.11). We find that few-shot prompts often yield similar results compared to simple

Table 2: **Baselines:** Results for baseline prompting strategies and models for Best@1 and Best@8.

| Method | Model | Best@1 | | | Best@8 | | |
|---|---|---|---|---|---|---|---|
| | | %Opt | Speedup | %Correct | %Opt | Speedup | %Correct |
| Instruction-Only | CODELLAMA 7B | 0.92% | 1.01× | 23.52% | 5.21% | 1.06× | 68.30% |
| Instruction-Only | CODELLAMA 13B | 0.41% | 1.00× | 10.02% | 2.45% | 1.03× | 40.49% |
| Instruction-Only | CODELLAMA 34B | 2.86% | 1.05× | 44.27% | 18.92% | 1.26× | 84.97% |
| Instruction-Only | GPT-3.5 | 16.26% | 1.20× | 80.67% | 39.16% | 1.54× | 98.77% |
| Instruction-Only | GPT-4 | 8.49% | 1.15× | **93.25%** | 21.17% | 1.31× | 98.77% |
| Few-Shot | CODELLAMA 7B | 2.15% | 1.02× | 43.46% | 9.51% | 1.15× | 85.07% |
| Few-Shot | CODELLAMA 13B | 2.25% | 1.02× | 40.29% | 13.70% | 1.21× | 83.03% |
| Few-Shot | CODELLAMA 34B | 2.66% | 1.02× | 43.97% | 13.70% | 1.16× | 82.62% |
| Few-Shot | GPT-3.5 | 11.45% | 1.13× | 80.98% | 29.04% | 1.38× | 95.91% |
| Few-Shot | GPT-4 | 18.92% | 1.25× | 82.82% | 36.40% | 1.44× | **98.98%** |
| COT | CODELLAMA 7B | 0.82% | 1.01× | 27.40% | 7.46% | 1.13× | 73.31% |
| COT | CODELLAMA 13B | 2.25% | 1.04× | 32.92% | 11.15% | 1.20× | 79.24% |
| COT | CODELLAMA 34B | 3.99% | 1.08× | 30.27% | 19.63% | 1.30× | 78.73% |
| COT | GPT-3.5 | 21.37% | 1.25× | 65.95% | **43.05%** | **1.60×** | 91.72% |
| COT | GPT-4 | **26.99%** | **1.32×** | 63.09% | 42.74% | 1.58× | 84.87% |

instruction-prompting. For instance, when prompted with instructions alone, both GPT-3.5 and CODEL-LAMA 34B demonstrated better %OPT and SPEEDUP metrics. This observation aligns with the findings of Zhao et al. (2021), which highlighted that few-shot examples can sometimes bias the model and lead to an incorrect understanding of the task. In the context of our study, the consistent use of the same fixed prompt might constrain the model to only apply optimization techniques present in the prompt, thereby resulting in sub-optimal performance. Finally, in line with the findings of Wei et al. (2022a) that identified CoT prompting as an emergent capability, we observe improvements with this approach over both instruction-tuned and fixed prompt setups, but notably only for the larger CODELLAMA (13B and 34B) and GPT-3.5 models. For CoT prompting; we note that GPT-4 outperforms GPT-3.5 Best@1 and under-performs GPT-3.5 Best@8: this may demonstrate a lack of output diversity from GPT4 despite using the same sampling hyper-parameters.

**Retrieval-based few-shot prompting.** Table 2 (bottom) shows results using our dynamic retrieval-based few-shot prompting strategy, with a preferable setting at $K = 4$ retrieved prompts. Extended results for $K \in \{1, 2, 4\}$ are detailed in Appendix A.7. The results show that dynamic few-shot prompting outperforms all the baseline variants, showing that PIE effectively adapts LLMs for program optimization in few-shot settings. We note that increased speedup may, however, come with some cost of correctness.

Table 3: **Dynamic retrieval-based few-shot prompting:** Results for dynamic retrieval-based few-shot prompting across models for Best@1 and Best@8.

| Method | Model | Best@1 | | | Best@8 | | |
|---|---|---|---|---|---|---|---|
| | | %Opt | Speedup | %Correct | %Opt | Speedup | %Correct |
| Dynamic Retrieval, K=4 | CODELLAMA 7B | 6.34% | 1.19× | 23.11% | 21.06% | 1.66× | 57.98% |
| Dynamic Retrieval, K=4 | CODELLAMA 13B | 9.30% | 1.29× | 26.99% | 28.12% | 2.04× | 62.58% |
| Dynamic Retrieval, K=4 | CODELLAMA 34B | 11.66% | 1.34× | 30.57% | 42.54% | 2.43× | 73.62% |
| Dynamic Retrieval, K=4 | GPT-3.5 | 28.02% | 1.55× | **79.65%** | 51.64% | 2.19× | **95.71%** |
| Dynamic Retrieval, K=4 | GPT-4 | **51.02%** | **2.53×** | 79.35% | **76.07%** | **3.93×** | **95.71%** |

## 4.2 RESULTS FOR FINETUNING

**Fine-tuning with PIE substantially improves all models.** We fine-tune CODELLAMA and GPT-3.5 models on our PIE dataset. Due to the cost of fine-tuning and sampling models through the OpenAI API, we were only able to train GPT-3.5 on the smaller, high-quality dataset (HQ) in Section 3.2. The top of Table 4 shows results for traditional fine-tuning on all models. We see substantially stronger results when fine-tuning on the smaller, high-quality dataset. These results reflect the observation that to adapt LLMs, a small set of high-quality examples can elicit strong performance (Zhou et al., 2023a; Chen et al., 2023).

**Performance-conditioned training outperforms fine-tuning.** Table 4 shows results for performance-conditioned (PERF-COND) generation (Section 3.2). Both fine-tuned CODELLAMA models (7B and 13B) show significant improvements in %OPT and SPEEDUP. These gains highlight how the performance improvement information (Figure 2) can enable models to distinguish optimal and sub-optimal solutions, leading to more effective optimizations.

**Synthetic data from self-play marginally improves generalization.** Next, we fine-tuned both CODELLAMA and GPT-3.5 using our PIE dataset augmented with our synthetic examples. We show results at the bottom of Table 4. For CODELLAMA and GPT-3.5, compared to using no synthetic data, the additional data improves both %OPT and often SPEEDUP, particularly with BEST@1. We believe the small set of synthetic examples helped generalize the fine-tuned model, as evidenced by the higher %OPT. [4]

**Model vs. Human Fastest Possible Speedup.** We compared the fastest human submissions in CodeNet for the test set with our model's fastest generation per problem. The comparison involved 39,129 model generations against 118,841 human accepted solutions (or 197,921 including incorrect or timed-out submissions). For the PIE dataset, the best human speedup achieved was 9.56×, while our model achieved a slightly higher speedup of 9.64×, setting a new upper limit.

Table 4: **Fine-Tuning:** Results for various models and dataset configurations.

| Dataset | Model | Best@1 | | | Best@8 | | |
|---|---|---|---|---|---|---|---|
| | | %Opt | Speedup | %Correct | %Opt | Speedup | %Correct |
| All | CODELLAMA 7B | 9.20% | 1.31× | 55.21% | 35.58% | 2.21× | 74.03% |
| All | CODELLAMA 13B | 12.78% | 1.52× | 55.42% | 43.76% | 2.71× | 75.46% |
| HQ | CODELLAMA 7B | 10.33% | 1.40× | **76.38%** | 45.30% | 3.14× | 87.63% |
| HQ | CODELLAMA 13B | 11.55% | 1.43× | 70.55% | 47.75% | 3.43× | 85.07% |
| HQ | GPT-3.5 | 38.55% | 2.70× | 59.10% | 86.71% | 6.74× | **95.40%** |
| All w/Perf-Cond | CODELLAMA 7B | 25.15% | 2.45× | 34.76% | 56.95% | 4.86× | 63.91% |
| All w/Perf-Cond | CODELLAMA 13B | 32.00% | **2.95×** | 38.55% | 66.56% | 5.65× | 70.96% |
| HQ + Self-Play | CODELLAMA 7B | 15.34% | 1.59× | 75.77% | 46.22% | 3.32× | 87.42% |
| HQ + Self-Play | CODELLAMA 13B | 14.31% | 1.61× | 76.28% | 49.69% | 3.51× | 86.20% |
| HQ + Self-Play | GPT-3.5 | **45.50%** | **3.02×** | 61.55% | **87.63%** | **6.86×** | 95.09% |

---

[4]For GPT-3.5, to be sure the increases came from the type of data and not the quantity of data, we performed an ablation by fine-tuning on the top 5,793 examples from PIE with a maximum of 8 duplicates (instead of the 5,570 pairs that included synthetic programs), and we saw BEST@1 performance degrade

### 4.3 DISCUSSION AND KEY TAKEAWAYS

**CODELLAMA vs. GPT-3.5 .** Our results demonstrate that openly available models such as CODELLAMA can be competitive with GPT-3.5. For prompting, CODELLAMA 34B with dynamic retrieval (42.54% %OPT, 2.43× SPEEDUP for BEST@8) roughly matched the performance of GPT-3.5 with dynamic retrieval (51.64% %OPT, 2.19× SPEEDUP for BEST@8). With fine-tuning, CODELLAMA 13B with performance-conditioned generation (66.56% %OPT, 5.65× SPEEDUP for BEST@8) approached the performance of GPT-3.5 with synthetic data (87.63% %OPT, 6.86× SPEEDUP for BEST@8); indeed, we may expect that fine-tuning CODELLAMA 34B using the same strategy would further bridge this gap. These results demonstrate that with the right adaptation strategies, open models can be competitive with private ones.

**Prompting vs. fine-tuning.** Our results demonstrate that while prompting can be an effective way to adapt models (with retrieval), fine-tuning significantly outperforms prompting for models of the same size.

**Effectiveness of retrieval-based few-shot learning.** Our results show that dynamic retrieval provides enormous gains over all other prompting approaches; for instance, it improved the performance of CODELLAMA 34B from 19.63 %OPT, 1.30× SPEEDUP to 34.25% %OPT, 2.28× SPEEDUP for BEST@8.

**Effectiveness of performance-conditioned generation.** We find that performance-conditioned generation is incredibly effective for achieving good performance; in particular, it improved the performance of CODEL-LAMA 13B from 47.75% %OPT, 3.43× SPEEDUP to 66.56% %OPT, 5.65× SPEEDUP for BEST@8.

## 5 RELATED WORK

Machine learning has been applied to improve performance by refactoring code (Mens & Tourwé, 2004; Agnihotri & Chug, 2020), identify compiler transformations (Bacon et al., 1994; Talaashrafi, 2022), perform parameter search (Hamadi, 2013; Huang et al., 2019; Kaufman et al., 2021; Seshadri et al., 2022; Kumar et al., 2022), auto-vectorize code (Nuzman et al., 2006; Mendis et al., 2019), optimize GPU code (Liou et al., 2020; Cummins et al., 2021), and automatically select algorithms (Kotthoff, 2016; Kerschke et al., 2019). and room at the top (Leiserson et al., 2020; Sherry & Thompson, 2021). DeepPERF (Garg et al., 2022) uses a transformer-based model fine-tuned to generate performance improvement patches for C# applications. Additionally, Chen et al. (2022) uses a discrete variational auto-encoder, each latent representation maps to a different category of code edits, and canonicalized code representations to automatically suggest performance improvements, Shypula et al. (2021) trains seq2seq models from scratch on optimization data to superoptimize assembly programs after compilation, Shi et al. (2019) trains tree-LSTM from scratch with RL to superoptimize halide IR, and MAGPIE (Blot & Petke, 2022) uses genetic algorithms for tasks including optimization. AlphaCode (Li et al., 2022) leverages language models to generate solutions to competitive programming problems in natural language, but it does not attempt to improve the performance of existing solutions. In contrast, we focus on adapting pre-trained LLMs (Chen et al., 2021b; Nijkamp et al., 2022; Tunstall et al., 2022; Xu et al., 2022; Fried et al., 2022) to performance optimization.

## 6 CONCLUSION

Our work is an initial step towards unlocking the potential of LLMs in leveraging the opportunities at the "top" of the computing stack. In particular, we improve algorithmic efficiency and, given a correctness oracle, enable automatic code optimization beyond optimizing compilers. Our results pave an exciting path for improving computing efficiency post Moore's law.

ACKNOWLEDGEMENTS

We extend our gratitude towards Herman Schmit, Chandu Thekkath, James Laudon, Cliff Young, and Stella Aslibekyan for reviewing the paper and providing insightful feedback. We also thank the extended team at Google DeepMind who enabled and supported this research direction. This material is partly based on research sponsored in part by the Air Force Research Laboratory (agreement number FA8750-19-2-0200 and award W911NF-20-1-0080). The U.S. Govt. is authorized to reproduce and distribute reprints for Governmental purposes notwithstanding any copyright notation thereon. The views and conclusions contained herein are those of the authors and should not be interpreted as necessarily representing the official policies or endorsements, either expressed or implied, of the Air Force Research Laboratory or the U.S. Government.

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

# A  APPENDIX

## A.1  ADDITIONAL ANALYSIS OF GENERATED CODE EDITS

We study the kinds of edits LLMs make that lead to our performance gains, focusing on our best-performing model, GPT-3.5 fine-tuned with synthetic data. We manually analyze a randomly sampled set of 120 (source, optimized) program pairs to understand the algorithmic and structural changes responsible for the performance gains. We find that the transformations can be broadly categorized into four kinds: *Algorithmic changes*, *Input/Output operations* (IO), *Data Structure modifications*, and *Miscellaneous adjustments*. *Algorithmic changes* (complex modifications, such as changing recursive methods to dynamic programming, and unexpected ones, such as omitting Binary Indexed Trees for simpler constructs) are most common, comprising ~34.15% of changes; *Input/Output operations* (e.g., changing 'cin/cout' to 'scanf/printf', efficiently reading strings) comprised ~26.02%; *Data Structures* (e.g., switching from vectors to arrays) comprised ~21.14%, and *Miscellaneous* (e.g., code cleanups and constant optimizations) comprised ~18.70%. These findings show the LLM's capability to perform sophisticated optimizations while preserving functionality.

In Appendix A.2, we show several examples to demonstrate the nature of optimizations made by our model. In these examples, we highlight the removal of a wasteful nested loop (Figure 4), eliminating the need to sort (Figure 3), avoiding unnecessary precomputations (Figure 5), use of simple modular arithmetic properties for optimization (Figure 6), and restructuring loops to improve performance (Figure 7).

**Algorithmic Transformations** ($34.15\%$). The most dominant transformation, representing approximately $34.15\%$ of the changes, is the *Algorithmic* category. Edits in this category exhibited sophisticated code restructuring. A frequent transformation was the shift from recursive methodologies to dynamic programming approaches, which can significantly enhance running time for specific problem types. Other examples include replacing Binary Indexed Trees with more straightforward constructs, removing redundant conditional checks, bit manipulations, and in some cases, using identities from number theory and algebra to replace complex computation with a formula.

**Input/Output Operations** ($26.02\%$). The *Input/Output operations* category, accounting for roughly $26.02\%$ of the changes, primarily centered on transitioning from C++ standard I/O methods ('cin/cout') to the faster C-standard methods ('scanf/printf'). Other examples include reading a string character-by-character vs. reading in one go, This transformation is particularly beneficial for problems dealing with extensive datasets, where I/O operations can be a bottleneck.

**Data Structure Modifications** ($21.14\%$). Changes in the *Data Structures* category, which constituted about $21.14\%$ of the transformations, showcased the model's adeptness in selecting optimal data structures for the task. A recurring modification was the transition from vectors to traditional arrays, leading to enhanced access times and reduced overhead. Additionally, the changes include removal of pointers in favor of direct access, and using hashmaps when appropriate.

**Miscellaneous Optimizations** ($18.70\%$). The *Miscellaneous* category, encompassing approximately $18.70\%$ of changes, captured a myriad of optimizations. These ranged from code cleanups, such as omitting unnecessary initializations, to replacing computationally intensive functions with predefined constants.

While our analysis showcases a variety of optimizations, it is essential to address certain speedup sources that may be considered spurious. Specifically, in 10 out of the 120 cases we examined, the speedup stemmed from reducing the constants used to allocate arrays. These speedups might not always reflect genuine algorithmic improvements, and indicate that the test cases may not perfectly cover all the cases, an open problem in code synthesis (Li et al., 2022). Thus, while they contribute to the overall speedup metrics, they should be interpreted with caution. Nevertheless, our analysis shows that the vast majority of speedups do not suffer from this issue, supporting our strong empirical results.

```cpp
int main(){
    int n, m, a, b;
    vector<int> v, v1;

    cin >> n >> m;

    for(int i = 0; i < m; i++){
        cin >> a >> b;
        v.push_back(a);
        v1.push_back(b);
    }

    sort(v.begin(), v.end());
    sort(v1.begin(), v1.end());

    if(v.back() > v1[0]){
        cout << 0 << endl;
    } else {
        cout << v1[0] - v.back() + 1 << endl;
    }

    return 0;
}
```

(a) Slower Code.

```cpp
int main(){
    int n, m, a, b, max = -1, min = 1e9;
    scanf("%d%d", &n, &m);
    for(int i = 0; i < m; i++){
        scanf("%d%d", &a, &b);
        if(a > max) max = a;
        if(b < min) min = b;
    }
    ans = min - max + 1;
    if(ans < 0){
        ans = 0;
    }
    printf("%d\n", ans);
    return 0;
}
```

(b) Faster Code.

Figure 3: Comparison of two programs for determining the range between the maximum and minimum values from a set of input pairs. The faster code (right) generated by PIE directly computes the maximum start and minimum end of the ranges in a single pass ($\mathcal{O}(n)$), eliminating the need for sorting ($\mathcal{O}(n \log n)$).

## A.2 EXAMPLES OF OPTIMIZATIONS

We show several examples to demonstrate the nature of optimizations made by our model. In these examples, we highlight the removal of a wasteful nested loop (Figure 4), eliminating the need to sort (Figure 3), avoiding unnecessary precomputations (Figure 5), use of simple modular arithmetic properties for optimization (Figure 6), and restructuring loops to improve performance (Figure 7).

```cpp
int main(){
    int k,x;
    cin>>k>>x;
    for (int i=-1000000;i<1000001;i++) {
        if(i==x){
            for (i=x-(k-1);i<x+k;i++){
                cout<< i<<" ";
            }
        }
    }
    return 0;
}
```

(a) Slower Code.

```cpp
int main(){
    int k,x;
    scanf("%d %d",&k,&x);
    for(int i=x-k+1;i<=x+k-1;i++)
        printf("%d ",i);
    return 0;
}
```

(b) Faster Code.

Figure 4: Comparison of two code implementations for printing $2k - 1$ consecutive numbers centered around the input $x$. The faster code (right) optimizes the process by directly computing the range without the need for nested loops, resulting in a more efficient and concise solution. The red highlighted portion in the slower code (left) indicates the wasteful nested loop that was eliminated in the optimized version. This loop unnecessarily iterates over a large range of numbers, only to perform a meaningful operation for a tiny fraction of those iterations.

```c
int main()
{
    int i, n;
    long long num[100005] = {0,1};
    for (i = 2; i <= 100004; i++)
        num[i] = (num[i-1] * i)%(1000000007);
    scanf("%d", &n);
    printf("%lld\n", num[n]);
    return 0;
}
```

(a) Slower Code.

```c
long long a=1,mod=1e9+7;
int n;
int main()
{
    scanf("%d",&n);
    for(int i=1;i<=n;i++)
    {
        a=(a*i)%mod;
    }
    printf("%lld",a);
}
```

(b) Faster Code.

Figure 5: Comparison of two code implementations for computing factorial modulo $10^9+7$. The slower code (left) precomputes the factorial for all numbers up to $10^5$, storing them in an array. The faster code (right) computes the factorial only for the given input, resulting in a more memory-efficient and faster solution. The red highlighted portion in the slower code indicates the precomputation step that was eliminated in the optimized version.

```c
int main() {
    int A, B, C;
    scanf("%d %d %d", &A, &B, &C);

    bool isYes = false;
    for (int i = 0; i < 1000; i++) {
        for (int j = 0; j < 1000; j++) {
            if ((A * i) - (B * j) == C)
                isYes = true;
        }
    }

    printf("%s\n", isYes ? "YES" : "NO");
    return 0;
}
```

(a) Slower Code with Nested Loops.

```c
int main() {
    int A, B, C;
    scanf("%d %d %d", &A, &B, &C);

    bool is_yes = false;
    for (int i = 0; i < B; i++) {
        if ((A * i) % B == C)
            is_yes = true;
    }

    printf("%s\n", is_yes ? "YES" : "NO");
    return 0;
}
```

(b) Optimized Code.

Figure 6: Optimization of a modular arithmetic problem. The slower code naively checks all possible combinations of i and j leading to a complexity of $\mathcal{O}(10^6)$. The faster code leverages the property of modular arithmetic, reducing the complexity to $\mathcal{O}(B)$. By directly computing the modulo operation for each i in the range $[0, B-1]$, it efficiently determines if the condition $(A \times i) \mod B = C$ is satisfied. Note that the example on the right is faster, but the generated code could have been even faster if it included a break statement.

```cpp
int main()
{
    int i,n,m;
    cin>>n>>m;
    for(i=m-n+1;i<m+n;i++){
        cout<<i;
        if(i!=m+n-1)
            cout<<" ";
    }
}
```

(a) Slower Code.

```cpp
int main(){
    int n,m;
    scanf("%d%d",&n,&m);
    for(int i=m-n+1;i<m;i++){
        printf("%d ",i);
    }
    printf("%d",m);
    for(int i=m+1;i<m+n;i++){
        printf(" %d",i);
    }
    printf("\n");
    return 0;
}
```

(b) Optimized Code.

Figure 7: Comparison of the slower code (left) with its optimized version (right). The optimized code avoids an additional conditional check inside the loop by restructuring the loop.

## A.3 CONVERGENCE OF GPT3.5 FINE-TUNED MODELS WITH ADDITIONAL GENERATIONS

The data in Section 4.2 shows that the gap between GPT-3.5 fine-tuned on HQ data and HQ + Self-Play seems to diminish with more generations. Training with HQ data helps increase the model's coverage, allowing it to optimize a large number of programs with just a single greedy sample. However, as more samples are drawn, the performance of HQ and HQ + Self-play gradually converge to a similar level of performance. We include the plots of the performance improvements as the number of samples gradually increases in Figure 8 and Figure 9.

Additionally, there is a slight drop in correctness after training with Self-Play; however, the speedup and correctness increase from 6.74 → 6.86 and 86.71 → 87.63. This reveals a precision-recall style trade-off: the model trained on synthetic data learns to try novel optimization strategies, but that comes at the cost of making more mistakes. We will add this analysis to the revision.

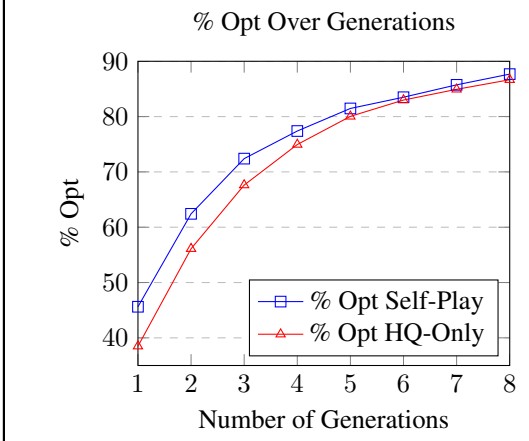

Figure 8: Performance in % Opt over generations comparison of GPT-3.5 fine-tuned with HQ Data Only vs. HQ + Self-Play.

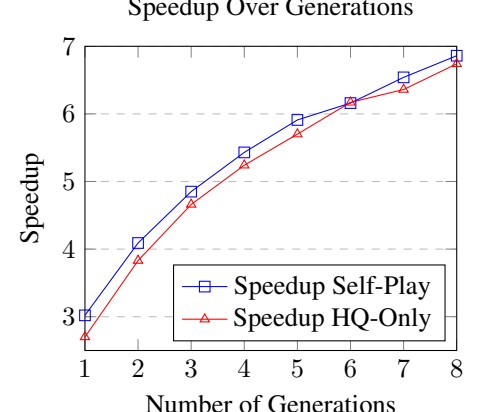

Figure 9: Speedup over generations comparison of GPT-3.5 fine-tuned with HQ Data Only vs. HQ + Self-Play.

Table 5: Error analysis of GPT-3.5 fine-tuned with synthetic data.

| Result | Percentage |
|---|---|
| Failed to compile (syntax/type errors) | 12.51% |
| Compiled, but got [95-100%] of test cases wrong | 27.59% |
| Compiled, but got (75, 95%] of test cases wrong | 12.08% |
| Compiled, but got (25, 75%] of test cases wrong | 10.84% |
| Compiled, but got (0-25%] of test cases wrong (at least 1 test case was wrong) | 6.90% |
| Ran all test cases, but the program was slower than the original | 9.93% |
| Ran all test cases, but the program was the same speed as the original | 9.40% |
| Ran all test cases, the program was faster, but not $1.1\times$ speedup or higher | 10.75% |

## A.4    ERROR ANALYSIS

We performed error analysis on the the GPT-3.5 fine-tuned with Self-Play. We analyzed the generated programs that it fails to optimize and the cause of each failure. Table 5 shows that a large fraction $\sim$60% of the failures happen because the proposed changes break a unit test. In about 30% of the cases, the model produces a correct, but the generated program is either slower (10%) or doesn't meet our threshold for speedup (10%). Finally, in about 10% of the cases, the generated program has a syntax error. Additionally, with test cases, we find that when the model gets the program wrong, it seems to most often get it *quite wrong* by missing most test cases.

Additionally, we conduct additional analysis to investigate the properties of programs that PIE fails to optimize. The results show a mild negative correlation between the problem description length and average accuracy (-0.15) and between the source program length and average accuracy (-0.26), suggesting longer inputs slightly reduce accuracy. Additionally, the average speedup has a mild negative correlation with both the problem description length (-0.16) and the source program length (-0.11), indicating a minimal impact of length on speedup compared to correctness. Overall, this analysis reveals that language models struggle to generate a correct program when faced with larger source programs and challenging problems, but their ability to optimize programs is minimally impacted. This motivates a future work direction where techniques from program repair may be combined with PIE for better results.

**Why does Performance-Conditioning Degrade the Ability to Produce Correct Code?**    We believe that conditioning the model only to generate programs with a 10/10 optimization rate may constrain the number of optimizations available for any given input. To investigate this, we experimented using the first 6 generations from the 7b Performance-Conditioned model when conditioned on 10/10 versus combining the first 2 generations when conditioned on 10/10, 9/10, and 8/10 (i.e. comparing 6 total generations from one strategy vs. 6 total generations across difference strategies). When we did this, we saw a %Correct increase from 59.95% to 64.36%. These results support the explanation that performance labels may restrict the set of generated programs that are correct.

## A.5 PIE DATASET DETAILS

| Dataset | Unique Problem IDs |
|---------|--------------------|
| Train   | 1,474              |
| Val     | 77                 |
| Test    | 41                 |

Table 6: Number of unique problem ids.

| Dataset | Pairs  |
|---------|--------|
| Train   | 77,967 |
| Val     | 2,544  |
| Test    | 978    |

Table 7: Number of pairs.

| Dataset | Mean src | Mean tgt | Median src | Median tgt |
|---------|----------|----------|------------|------------|
| Train   | 675.00   | 616.44   | 417        | 372        |
| Val     | 644.74   | 471.47   | 180        | 110        |
| Test    | 427.85   | 399.15   | 362        | 319        |

Table 8: GPT-2 Tokenizer lengths.

## A.6 SELF-PLAY DATA GENERATION DETAILS

We use the template in Figure 10 for prompting GPT-3.5 in the self-play scenario. For the prompt, we sample natural language descriptions of programming problems as well as accepted solutions to fill in the template. For generation, we use a temperature of 1.0 and use top-p sampling with $p = 0.9$ For each prompt, we try attempt to take $n = 5$ samples. We chose these samples after doing a sweep of 6 configurations of generation parameters, each attempting to generate 200 programs. We found this configuration to be the most cost-effective per new-sample with relatively promising rates of novelty.

We found that after attempting to generate 10,000 new programs through the prompting strategy, 6,553 were not in the training/validation/test set of PIE. We keep track of equivalent programs of the ones generated, and of these 6,553 generations we found 3,314 equivalence sets. In total, this required executing over 1.4 million binary input pairs. Parallelized on a 24-core Intel 13900k processor with 64GB of RAM, this took less than 72 hours to complete.

## A.7 ABLATION OF RETRIEVAL-BASED FEW-SHOT PROMPTING CONFIGURATION

For our retrieval-based prompting experiment we tried multiple configurations for the number of retrieved prompts where of $K = \{1, 2, 4\}$ of the $K$ closest retrieved prompts.

## A.8 TRAINING DETAILS

We fine-tuned the 7B and 13B variants using the HuggingFace Transformers library with FSDP to distribute the training process across $8\times$ 48GB GPUs (NVIDIA RTX A6000/NVIDIA L40). For our high-quality

```
Description 1: {description_1}
Code 1: {code_1}
Description 2: {description_2}
Code 2: {code_2}
Now, can you generate a program that takes that same input as Code 2
↪   in Code 3 but produces different outputs? Write it to be as novel
↪   as possible.
Code 3:
```

Figure 10: The prompt template used for prompting GPT-3.5 for generating synthetic data for self-play.

Table 9: Retrieval-based few-shot prompting ablation over different $K$ examples for retrieval and over various models.

| Method | Model | Best@1 | | | Best@8 | | |
|---|---|---|---|---|---|---|---|
| | | %Opt | Speedup | %Correct | %Opt | Speedup | %Correct |
| Dynamic Retrieval, K=1 | CODELLAMA 7B | 3.27% | 1.09× | 16.67% | 15.64% | 1.50× | 50.51% |
| Dynamic Retrieval, K=1 | CODELLAMA 13B | 5.32% | 1.16× | 21.68% | 22.29% | 1.72× | 62.99% |
| Dynamic Retrieval, K=1 | CODELLAMA 34B | 10.02% | 1.25× | 30.67% | 34.25% | 2.21× | 69.73% |
| Dynamic Retrieval, K=2 | CODELLAMA 7B | 4.40% | 1.13× | 20.55% | 16.87% | 1.51× | 55.32% |
| Dynamic Retrieval, K=2 | CODELLAMA 13B | 9.10% | 1.35× | 28.73% | 28.02% | 1.97× | 64.72% |
| Dynamic Retrieval, K=2 | CODELLAMA 34B | 10.22% | 1.27× | 25.87% | 34.25% | 2.28× | 63.19% |
| Dynamic Retrieval, K=2 | GPT3.5 | 26.18% | 1.58× | 80.37% | 48.06% | 2.14× | **97.85%** |
| Dynamic Retrieval, K=2 | GPT4 | 50.00% | **2.61×** | **80.57%** | 74.74% | 3.95× | 97.85% |
| Dynamic Retrieval, K=4 | CODELLAMA 7B | 6.34% | 1.19× | 23.11% | 21.06% | 1.66× | 57.98% |
| Dynamic Retrieval, K=4 | CODELLAMA 13B | 9.30% | 1.29× | 26.99% | 28.12% | 2.04× | 62.58% |
| Dynamic Retrieval, K=4 | CODELLAMA 34B | 11.66% | 1.34× | 30.57% | 42.54% | 2.43× | 73.62% |
| Dynamic Retrieval, K=4 | GPT-3.5 | 28.02% | 1.55× | 79.65% | 51.64% | 2.19× | 95.71% |
| Dynamic Retrieval, K=4 | GPT-4 | **51.02%** | 2.53× | 79.35% | **76.07%** | **3.93×** | 95.71% |

dataset, which consists of approximately 4,000 examples, the models were fine-tuned until convergence was achieved, which can be done under 12 hours with 8 GPUs. For tasks related to full data fine-tuning and performance-conditioned fine-tuning, we only train for 1 epoch, which takes 24 to 36 hours, depending on the model of GPU used. All experiments were conducted using the AdamW optimizer (Loshchilov & Hutter, 2017). For the 7B and 13B variants of CODELLAMA, we used a batch size of 32 and a learning rate of $1e-5$ for all of the experiments.

### A.9 EXAMPLE OF DUPLICATE CODE IN CODENET WITH DIFFERENT MEASURED RUNTIMES

Figure 11 contains an example of code we found duplicated across the Project Codenet Dataset with variance in the dataset's report of CPUTime. For problem number p03160 and between submission s766827701 and s964782197 a speedup of 2.44× is reported, despite the programs and environments being identical. We note that multiple submissions existed, because it was template code. For brevity, we remove the macros, imports, and comments.

```cpp
using namespace std;
typedef long long ll;
inline void getInt(int* p);
const int maxn=1000010;
const int inf=0x3f3f3f3f;
ll n;
ll dp[maxn];
ll a[maxn];
int main()
{
    gbtb;
    cin>>n;
    repd(i,1,n)
    {
        cin>>a[i];
    }
    dp[1]=0;
    dp[0]=0;
    dp[2]=abs(a[2]-a[1]);
    repd(i,3,n)
    {
        dp[i]=min(dp[i-2]+abs(a[i]-a[i-2]),dp[i-1]+abs(a[i]-a[i-1]));

    }
    cout<<dp[n];
    return 0;
}

inline void getInt(int* p) {
    char ch;
    do {
        ch = getchar();
    } while (ch == ' '  ch == '\n');
    if (ch == '-') {
        *p = -(getchar() - '0');
        while ((ch = getchar()) >= '0' && ch <= '9') {
            *p = *p * 10 - ch + '0';
        }
    }
    else {
        *p = ch - '0';
        while ((ch = getchar()) >= '0' && ch <= '9') {
            *p = *p * 10 + ch - '0';
        }
    }
}
```

Figure 11: An example of a C++ program we found multiple submissions for as it is template code. Across these submissions, we found variance in the reported CPU runtime despite the code and competitive programming environment being identical.

### A.10 LORA RESULTS

We show results using low-rank adaptors for finetuning in Table 10. We hypothesize that this gap may be because performance optimization examples do not occur naturally in the training data.

Recent work has shown that the effectiveness of parameter-efficient methods depends on the training data. For example, He et al. (2021) find that "PEFT techniques are slower to converge than full tuning in low/medium-resource scenarios," and Niederfahrenhorst et al. (2023) find that LoRA is least effective for challenging tasks like mathematical reasoning. Together, these works indicate that the performance of PEFT may be heavily task-dependent. Our hypothesis is based on the fact that LoRA only changes a small subset of the model's parameters, and is likely most helpful when the base model has some proficiency for the task (due to pre-training), and LoRA can help adapt the model of the task further. Given that LLMs generally struggled in program optimization without retrieval or full fine-tuning, we hypothesize that the challenging nature of the problem and a potential lack of pre-trained proficiency pose challenges for LoRA.

Table 10: LoRA Experiments: Results for fine-tuning CODELLAMA with low rank adapters. A LoRA rank of 32 and LoRA alpha of 16 is used for all experiments listed.

| Dataset | Model | Best@1 | | | Best@8 | | |
|---|---|---|---|---|---|---|---|
| | | %Opt | Speedup | %Correct | %Opt | Speedup | %Correct |
| All | CODELLAMA 7B | **1.12**% | 1.01× | 45.82% | 9.57% | **1.17**× | 87.47% |
| All | CODELLAMA 13B | 0.41% | 1.01× | 59.47% | 9.67% | 1.15× | 90.94% |
| HQ | CODELLAMA 13B | 0.92% | **1.02**× | **59.57**% | **10.69**% | **1.17**× | **91.04**% |

### A.11 PROMPTS

```
Given the program below, improve its performance:

### Program:
{src_code}

### Optimized Version:
```

Figure 12: Instruction-prompting for adapting LLMs. The model is provided with direct instructions to improve the performance of the given program.

```
slow1 → fast1 || slow2 → fast2 || slow3 → fast3 || ... || slowN →
↪  fastN

### Program:
{src_code}

### Optimized Version:
```

Figure 13: Few-shot prompting for in-context learning. The format "slow → fast" is used for adaptation. A test program is appended for inference.

```
### slower program:
{src_code}
Give a potential strategy improving the program:
```
LLM reasoning: The current program calculates the sum of integers from 1 to n-1 using a loop. This iterative approach results in a time complexity of O(n). The sum of the first n natural numbers follows an arithmetic progression, and by leveraging Gauss's formula, n(n-1)/2, we can compute the sum directly. This formula-based approach allows us to calculate the sum in constant time, O(1), eliminating the need for iteration.
```
### optimized version of the same code with the strategy mentioned
↪  above:
```
LLM Response: optimized code

Figure 14: Chain-of-thought prompting. The model's intermediate response and final program are highlighted in blue, indicating they are produced by the LLM.

```
similar_slow1 → similar_fast1 || similar_slow2 → similar_fast2 || ...
↪  || similar_slowN → similar_fastN

### Program:
{src_code}

### Optimized Version:
```

Figure 15: Retrieval-based few-shot prompting. By dynamically retrieving analogous program structures or challenges, the model is guided to better harness patterns in PIE.

```
Below is a program. Optimize the
↪   program and provide a more
↪   efficient version.

### Program:
{src_code}

### Optimized Version:
{fast_code}
```

(a) Training Prompt.

```
Below is a program. Optimize the
↪   program and provide a more
↪   efficient version.

### Program:
{src_code}

### Optimized Version:
```

(b) Inference Prompt.

Figure 16: Standard training and inference prompts with PIE.

```cpp
// Retrieved 1-nearest prompt, slower
    src_code
#include <iostream>
#include <stack>
using namespace std;

stack<char> s;

int main() {
    int n;
    cin >> n;
    for (int i = 0; i < n; ++i) {
        char t;
        cin >> t;
        if (s.empty())
            s.push(t);
        else if (t == s.top())
            ;
        else
            s.push(t);
    }
    cout << s.size();
    return 0;
}
```

(a) Retrieved Slow.

```cpp
// Retrieved 1-nearest prompt, faster
    tgt_code
#include <cstdio>

int n, ans;
char ch1, ch2;

int main() {
    scanf("%d", &n);
    ch1 = getchar();
    ch1 = getchar();
    ans = 1;
    for (int i = 1; i < n; i++) {
        ch2 = getchar();
        if (ch2 != ch1) ans++;
        ch1 = ch2;
    }
    printf("%d", ans);
}
```

(b) Retrieved Fast.

```cpp
// Code for to_be_optimized goes here
#include <bits/stdc++.h>
using namespace std;

int main() {
    int N, len;
    cin >> N;
    string s;
    cin >> s;
    len = s.size();
    if (len > N) {
        for (int i = len; i > N; i--) {
            s.pop_back();
        }
        for (int j = 0; j < 3; j++) {
            s.push_back('.');
        }
        cout << s;
    } else {
        cout << s;
    }
    return 0;
}
```

(c) Program to be optimized.

Figure 17: Example of retrieval-based prompting. To optimized the program in Figure 17(c), our dynamic prompting method retrieves the closest source program from the training set (Figure 17(a)), where the similarity is measured using CodeBertScore (Zhou et al., 2023b). The slow program and the corresponding fast program (Figure 17(b)) from the training set are used as prompts.

