# OpenReview forum: "Learning Performance-Improving Code Edits"
_ICLR.cc/2024/Conference — ICLR 2024 spotlight_

### Official Review · Reviewer_jit6 · 2023-10-30

**Soundness:** 4 excellent
**Presentation:** 4 excellent
**Contribution:** 3 good
**Rating:** 8
**Confidence:** 4

**Summary:**

This paper introduces a dataset for learning code performance improvements. Based on this dataset, the capabilities of both open-source (CodeLlama) and proprietary models (GPT3.5/GPT4) to optimize code were tested using various PROMOTING and FINETUNING methods. The effectiveness of different methods and the performance differences between open-source and proprietary models are analyzed based on the experimental data.

**Strengths:**

This paper conducts comprehensive experiments on the task of improving code performance using LLM, and offers a comparative analysis of the effectiveness of various recent methods. This work will contribute to aiding researchers in the domain to better enhance the capabilities of code generation models.

**Weaknesses:**

1. Regarding the experiment in Section 4.1: In Table 2, the performance of GPT-4 under the Few-Shot and CoT methods for code optimization was not tested.

2. Data in Section 4.2 shows that after training on the synthetic dataset, there is a noticeable performance improvement in Best@1, but only a slight increase in Best@8. In fact, for the combination of < HQ+Self-Play, GPT-3.5 >, there is a minor performance decrease (95.42%->95.11%). The paper does not provide a convincing explanation for this.

3. Different types of code have varying scopes and complexities for optimization. The paper does not clearly state the proportion of each language type in the training data nor in the test code.

**Questions:**

1. How do you view the impact of training LLMs using synthetic data (generated either by the trained model itself or from other models)? Assuming synthetic data is generated by the trained model M and undergoes data filtering, how would the output distribution of M change?

2. I noticed a statement regarding the comparison of LoRA training: "We hypothesize that this gap may be because performance optimization examples do not occur naturally in the training data." Can you provide a clearer version of this statement, or offer a more specific explanation?

---

> ### Author Response · Authors · 2023-11-17
>
> Thank you for taking the time to understand the paper and provide feedback. We're glad to hear you feel the experiments are comprehensive and sound, and we're happy that you think our work will be beneficial to the community.
> We've provided our responses below, and we're happy to address additional questions if anything is unclear!
>
> ---
>
> ### **[R4-Q1] How do you view the impact of training LLMs using synthetic data?**
>
> The last three rows of fine-tuning results in Table 3 show the result of fine-tuning on synthetic data generated by GPT3.5 (in addition to data from PIE); we see increased generalization. The synthetic data was generated by the pre-trained GPT3.5 turbo model, and the optimized programs were then generated by GPT3.5 fine-tuned on 4K high-quality pairs. GPT3.5-turbo fine-tuned on the PIE dataset. Generally, training with this synthetic data enhances generalization ability. Please let us know if this does not answer your question, and we will be happy to provide additional clarifications.
>
> ---
>
> ### **[R4-Q2] Can you provide a clearer version of the statement regarding LoRA?**
>
> We hypothesize that this gap may be because performance optimization examples do not occur naturally in the training data.
>
> Recent work has shown that the effectiveness of parameter-efficient methods depends on the training data. For example, [1] find that "PEFT techniques are slower to converge than full tuning in low/medium-resource scenarios," and [2] find that LoRA is least effective for challenging tasks like mathematical reasoning. Together, these works indicate that the performance of PEFT may be heavily task-dependent. Our hypothesis is based on the fact that LoRA only changes a small subset of the model's parameters, and is likely most helpful when the base model has some proficiency for the task (due to pre-training), and LoRA can help "adapt" the model of the task further. Given that LLMs generally struggled in program optimization without retrieval or full fine-tuning, we hypothesize that the challenging nature of the problem and a potential lack of pre-trained proficiency pose challenges for LoRA. We will add these new citations in the next version and clarify this point.
> [1] He, Junxian, Chunting Zhou, Xuezhe Ma, Taylor Berg-Kirkpatrick, and Graham Neubig. "Towards a unified view of parameter-efficient transfer learning." arXiv preprint arXiv:2110.04366 (2021).
> [2] Fine-Tuning LLMs: LoRA or Full-Parameter? An in-depth Analysis with Llama 2
>
> ---
>
> ### **[R4-Q3] The Performance of GPT-4 under Few-Shot and CoT were not tested**
>
> At the time of writing, we did not have the funds to run all experiments with GPT-4. We were able to get the funds, and we have evaluated the outputs, and we will include them in the draft. For GPT4, both Few Shot and CoT prompting improve over Instruction prompting but underperform retrieval-based prompting. For example, Best@4 for CoT has a %Opt of 38.60% and a speedup of 1.51$\times$, while Few-Shot has a %Opt of 30.83% and a speedup of 1.37$\times$.
>
>
> | Prompting Method | %Opt | Speedup |
> |------------------|--------|---------|
> | Instruction Only (Best@4) | 16.58% | 1.23$\times$|
> | CoT (Best@4) | 38.60% | 1.51$\times$ |
> | Few-Shot (Best@4) | 30.83% | 1.37$\times$ |
> | Dynamic Retrieval (Best@4) | 69.03% | 3.56$\times$|
>
>
> ---
>
> ### **[R4-Q4] Different types of code have varying scopes and complexities for optimization. The paper does not clearly state the proportion of each language type in the training data nor in the test code.**
>
> Our training/validation/test sets consist entirely of C++ programs. We focus on C++ since it is a performance-oriented language compatible with the gem5 simulator. We will update the draft to make this more clear.
>
> ---
> ---
>
> ## [Go to Reviews](https://openreview.net/forum?id=ix7rLVHXyY&noteId=zNxhW6pwmc)
>
> ---
> ---

---

> ### Author Response · Authors · 2023-11-17
>
> ### **[R4-Q5]  Data in Section 4.2 shows that after training on the synthetic dataset, there is a noticeable performance improvement in Best@1 (38.49 → 45.62, 2.70 → 3.02), but only a slight increase in Best@8 (86.66 →87.68, 6.74→6.86). In fact, for the combination of < HQ+Self-Play, GPT-3.5 >, there is a minor performance decrease (95.42%→95.11%).**
>
> Training with HQ data helps increase the model's coverage, allowing it to optimize a large number of programs with just a single greedy sample. However, as more samples are drawn, both HQ and HQ + Self-play gradually converge to a similar level of performance. We verify this by computing the performance improvements as the number of samples gradually increases:
>
>
> | Number of Generations | % Opt Self-Play |  % Opt  HQ-Only | Diff  |
> |-----------------------|---------|---------|--------|
> | 1                     | 45.62%            | 38.49% | 7.13% |
> | 2                     | 62.42%            | 56.11% | 6.31% |
> | 3                     | 72.40%            | 67.62% | 4.79% |
> | 4                     | 77.39%            | 74.95% | 2.44% |
> | 5                     | 81.47%            | 80.04% | 1.43% |
> | 6                     | 83.50%            | 82.99% | 0.51% |
> | 7                     | 85.74%            | 84.93% | 0.81% |
> | 8                     | 87.68%            | 86.66% | 1.02% |
>
>
> | Number of Generations | Speedup Self-Play |  Speedup HQ-Only | Diff |
> |-----------------------|---------|-------------------|------|
> | 1                     | 3.02    | 2.70              | 0.32 |
> | 2                     | 4.09    | 3.83              | 0.26 |
> | 3                     | 4.85    | 4.66              | 0.19 |
> | 4                     | 5.43    | 5.24              | 0.19 |
> | 5                     | 5.91    | 5.70              | 0.21 |
> | 6                     | 6.16    | 6.17              | -0.01|
> | 7                     | 6.54    | 6.36              | 0.17 |
> | 8                     | 6.86    | 6.74              | 0.12 |
>
> ---
>
> ### **[R4-Q6] In fact, for the combination of < HQ+Self-Play, GPT-3.5 >, there is a minor performance decrease (95.42%$\rightarrow$95.11%)**
>
> Despite the slight drop in correctness, the speedup and correctness increase from 6.74 → 6.86 and 86.66 → 87.68. This reveals a precision-recall style tradeoff: the model trained on synthetic data learns to try novel optimization strategies, but that comes at the cost of making more mistakes. We will add this analysis to the revision.
>
> ---
> ---
>
> ## [Go to Reviews](https://openreview.net/forum?id=ix7rLVHXyY&noteId=zNxhW6pwmc)
>
> ---
> ---

---

> ### Author Response · Authors · 2023-11-22
>
> We hope that you are satisfied with our answers and the additional results we have provided. As the discussion period comes to an end, we would be grateful if you could let us know if we have adequately addressed your comments and if you have any further questions.

---

> > ### Author Response · Authors · 2023-11-23
> > **Thanks for your time and feedback!**
> >
> > Dear reviewer jit6,
> >
> > Thanks for your time and feedback, which will help us improve our work. We hope that we were able to address your concerns.
> >
> > Best,
> >
> > Authors

---

### Official Review · Reviewer_VFMY · 2023-10-31

**Soundness:** 2 fair
**Presentation:** 3 good
**Contribution:** 2 fair
**Rating:** 5
**Confidence:** 4

**Summary:**

This paper proposes a dataset called Performance-Improving Edits (PIE). It contains 77K pairs of C++ programs where one is a performance-improved version of the other, filtered from CodeNet. It also proposes to use gem5 simulator to simulate an Intel CPU as the runtime metric rather than running the programs and measuring wall time. The paper proposes a number of ways to use the PIE dataset to induce large language models to improve code performance, including prompt engineering, retrieval augmented generation, and fine-tuning. The best result is from fine-tuning GPT-3.5, with a surprising speedup of 6.86X, beating best human performance.

**Strengths:**

This work is well motivated and addresses a meaningful task. The construction process described in section 2 seems reasonable. The different methods to adapt models in section 3 cover most main stream methods.

**Weaknesses:**

The results are too good to be true. It's surprising to see that, for C++ competitive programming tasks, the fine-tuned GPT-3.5 beats the best human submission by a large margin: 6.86X speed up versus 4.06X speed up. So let's take a look at the examples in appendix A.1 which are code improvements generated by the model. Figure 3 of A.1 contains two programs that are functionally different. Figure 4(a) in A.1 is so bad that it seems unlikely in a C++ competition. Figure 5 contains two programs that are functionally different and also have different interfaces for execution (unclear how it passes correctness test). Figure 6 contains two programs that are functionally different and Figure 6(b) is missing an obvious break statement to help performance. Overall none of before-optimization examples seems plausible as competitive C++ programming submissions. By looking at these examples, I question the quality of the PIE dataset.

I randomly looked at some entries in the supplementary material and they seem consistent with examples in appendix A.1 and not plausible for competitive C++ programming submissions.

A side evidence for the issue can also be seen in Tables 1 and 2, where the correct percentages are high and indicate that the difficulty level of the programming tasks is low.

It is stated on page 6 that "For generations that are either incorrect or slower than the original program, we use a speedup of 1.0 for that example, given that, in the worst case, the original program has a speedup of 1.0." This explains some of the 6.86X result: the evaluation setup is such that only good generations are considered in calculating the metric.

**Questions:**

The following are questions in addition to the weakness section.

The rationale of using gem5 simulation instead of measuring wall time makes sense but won't this run the risk of overly focused on a single CPU (Inter Skylake)? Also gem5 simulation still does not capture big O complexity which is often a goal in optimizing code.

It is counter-intuitive that Section 3.2 only uses 4K out of the 77K pairs in PIE to fine-tune GPT 3.5. Then 3K synthetic data are created and used. Could you explain the rationale of these choices? This seems to contradict the quality claim of the PIE dataset.

---

> ### Author Response · Authors · 2023-11-17
>
> Thank you for taking the time to review our paper. We were happy to read that you find PIE well motivated and its construction process reasonable, and appreciated the coverage of our experiments. We think that all your questions are addressable within this discussion period. Please see our response below. We would be more than happy to address additional questions during the discussion period if anything remains unclear.
>
> ---
>
> > ### **[R3-Q1] Will using gem5 risk being overly-focused on a single CPU (Intel Skylake)?**
>
> In Section 4, we analyzed 120 randomly selected programs in the test set, and all optimizations seemed generic with nothing specific to x86, let alone Skylake. Generally all optimizations we observed would execute faster on any backend: for example, using a faster sorting algorithm will almost always be faster on any backend.
>
> We went further to check this systematically. To our knowledge the most plausible ways to customize C++ for a certain backend is through direct writing of assembly, SIMD/vector programming, and pragma statements (e.g. to specify memory prefetching). For the GPT3.5 model trained with self-play, **none** of the solutions contained assembly or vector programing, and only 0.3% of optimized generations contained pragmas that could affect execution time. It is intuitive that the model doesn’t learn this behavior, because programers doing program competitions seem to rarely focus on optimizing for the program website backend.
>
> ---
>
> > ### **[R3-Q2] Also gem5 simulation still does not capture big O complexity which is often a goal in optimizing code.**
>
> Big O complexities are impossible to automatically infer in the general case, and thus cannot be the basis for an automatic metric. Furthermore, Big O captures *asymptotic* complexities and can hide large constants. By measuring actual runtime on provided test cases, our evaluation best aligns with how user submissions are evaluated in practice.
>
> ---
>
> > ### **[R3-Q3] Why do we only use 4K programs in section 3.2 and add more synthetic programs later, instead of real programs.**
>
> As described in Section 3.2, the reason we did not finetune GPT-3.5-turbo on our full PIE dataset is due to our limited API budget. At a high level, our results in Table 3 demonstrate the following for the CodeLlama models:
>
> - Performance-conditioned models trained on PIE perform the best (by a wide margin)
> - Models trained on the 4K "high quality" subset of PIE perform second best
> - Models trained on all of PIE (without performance conditioning) are third
>
> These results make sense, because finetuning on all of PIE without performance conditioning encourages the model to generate modest speedups. We ideally would have finetuned GPT-3.5-turbo using performance conditioning on the full PIE dataset, but we did not have the budget to do so. Thus, we used the second best option, which was to train GPT-3.5-turbo on the high quality examples, which was also significantly cheaper (~20x cheaper).
>
> The synthetic data was introduced to solve a separate issue, which is the limited number of problems in the dataset. Even if we have a large number of submissions, they are all for the same 1474 problems. Our synthetic data consists of *new problems*, which helps our techniques generalize better to new problems (recall that our train/test split is at the level of problems, not code submissions). Because of this, synthetic data helps improve performance compared to just using PIE. On the page 8 footnote we report an ablation demonstrating the gains are from the type of data added, not the quantity of data.
>
> ---
> ---
>
> ## [Go to Reviews](https://openreview.net/forum?id=ix7rLVHXyY&noteId=vLeyg6r8nW)
>
> ---
> ---

---

> ### Author Response · Authors · 2023-11-17
>
> > ### **[R3-Q4] The results are too good to be true. It's surprising to see that the fine-tuned GPT-3.5 beats the best human submission by a large margin: 6.86$\times$ speed up versus 4.06$\times$ speed up.**
>
> We do not believe that our results are surprising or too good to be true. First, all examples in our dataset (including the ones in our paper) are from CodeNet, a widely used dataset in code generation and program synthesis [1, 2, 3]. Importantly, CodeNet contains a wide variety of submissions, including early drafts as well as submissions from novice programmers. This explains why some of the programs are slow or lower quality. In this light, we believe the 6.86$\times$ speedup we obtain is highly plausible. Furthermore, we note that the 6.86$\times$ speedup is only possible when using multiple samples. With only one sample, over 54% of the test set is not optimized more than 1.10$\times$ and the mean speedup is 3.02$\times$.
>
>
> [1] Guo, Daya, Shuai Lu, Nan Duan, Yanlin Wang, Ming Zhou, and Jian Yin. "[Unixcoder: Unified Cross-modal Pre-training for Code Representation](https://arxiv.org/abs/2203.03850)." arXiv preprint arXiv:2203.03850 (2022).
>
> [2] Li, Yujia, David Choi, Junyoung Chung, Nate Kushman, Julian Schrittwieser, Rémi Leblond, Tom Eccles et al. "[Competition-level Code Generation with AlphaCode](https://arxiv.org/abs/2203.07814)" Science 378, no. 6624 (2022): 1092-1097.
>
> [3] Nye, Maxwell, Anders Johan Andreassen, Guy Gur-Ari, Henryk Michalewski, Jacob Austin, David Bieber, David Dohan et al. "[Show your work: Scratchpads for Intermediate Computation with Language Models](https://arxiv.org/abs/2112.00114)" arXiv preprint arXiv:2112.00114 (2021).
>
> ---
>
> > ### **[R3-Q5] High correctness implies that the difficulty of programming tasks is low**
>
> We created the “slow”-”fast” example pairs in our dataset by taking **only accepted submissions** that pass all test cases. Because the input is already correct, a model that only copies the input verbatim is 100% correct. Thus, high correctness in isolation is trivial to achieve. In contrast, %Opt and Speedup (which both require all test cases to pass) can be difficult; especially with formulaic prompting strategies. We included correctness to help with analysis and understanding of general model performance characteristics.
>
> ---
>
> > ### **[R3-Q6] Examples in the appendix are “functionally different”**
>
> All three of these functions pass all of our unit tests described in Section 2. Moreover, we exhaustively tested all three based on the accepted range of inputs from the problem description: all three program pairs are equivalent.
>
> The problems in our benchmark come with input ranges, and programs are only required to provide the correct output on inputs inside these ranges; we will clarify this point in our paper. We give details below; we are also happy to share the scripts that we used to exhaustively test the problems.
>
> Regarding the broader issue of how we check correctness, we emphasize that the use of unit tests to check correctness is the standard way to check correctness in competitive programming, and is also standard in the program synthesis literature (under the name “programming by example”); indeed, example-based synthesis has been integrated into commercial tools such as the FlashFill tool for Excel [1]. We would also like to note that CodeNet by default comes with only 5 test cases per program, and we increased the coverage to a median of over 100 test cases per program by augmenting additional test cases from Alphacode (Section 2).
>
> [1] Sumit Gulwani, [Automating String Processing in Spreadsheets using Input-Output Examples](https://dl.acm.org/doi/abs/10.1145/1925844.1926423?casa_token=_p-wn3-qwpwAAAAA:lMSWSPoREj_71zl3Di3vuq79-eY67y2F-tEzu1L72gXWmtWfjw07lGkVgrBw2mKbXHRtCoInN79IUw). In POPL, 2011.
>
> ---
> ---
>
> ## [Go to Reviews](https://openreview.net/forum?id=ix7rLVHXyY&noteId=vLeyg6r8nW)
>
> ---
> ---

---

> ### Author Response · Authors · 2023-11-17
>
> > ### **[R3-Q7] “Figure 3 of A.1 contains two programs that are functionally different”**
>
> Problem URL: https://atcoder.jp/contests/abc127/tasks/abc127_c
>
> Input Space: $$1 <= N <= 1e5; 1 <= M <= 1e5; 1 <= L_i <= R_i <= N$$
>
> In addition to ~100 unit tests, we randomly generated 10,000 tests within the provided input bounds, and found that the two programs produced equal outputs on all of these inputs. Both programs aim to find the range between the maximum of the first set of elements and the minimum of the second set of elements. If the maximum of the first elements is greater than or equal to the minimum of the second elements, they both output 0. Otherwise, they output the difference plus 1. The first program sorts two vectors containing all $a$s and $b$s to find these values, while the second program directly computes the maximum $a$ and minimum $b$ during the input phase. However, the second program is more efficient, as it operates in linear time, avoiding the sorting overhead present in the first program.
>
> ---
>
> > ### **[R3-Q8] “Figure 5 contains two programs that are functionally different and also have different interfaces for execution (unclear how it passes correctness test)”**
>
> Problem URL: https://atcoder.jp/contests/abc055/tasks/abc055_b
>
> Input Space: $$1 <= N <= 1e5$$
>
> After brute-force enumerating over all possible inputs in the specified input range, both programs are equivalent. Both programs calculate the factorial of an integer $n$ modulo $10^9 + 7$, but the second program is significantly faster, as it eliminates the overhead of precalculation and array storage, resulting in a faster execution for each individual input of $n$.
>
> ---
>
> > ### **[R3-Q9] Figure 6 contains two programs that are functionally different**
>
> Problem URL: https://atcoder.jp/contests/abc060/tasks/abc060_b
>
> Input Space: $$1<= A <= 100; 1 <= B <= 100; 0 <= C < B$$
>
> After brute-force enumerating over all possible inputs in the specified input range, both programs were equivalent. Both programs determine if an integer solution exists for the equation $(A \times i) - (B \times j) = C$, but the second program is significantly more efficient by leveraging modular arithmetic, reducing the problem to a single loop with a maximum of $B$ iterations, compared to the first program's exhaustive search with nested loops.
>
> ---
>
> > ### **[R3-Q10] The submission in Figure 4(a) in A.1 seems unlikely from a C++ competition**
>
> The submissions are all from CodeNet (Puri et al., 2021). The dataset obtained from the [AtCoder](https://atcoder.jp/) or [AIZU](https://judge.u-aizu.ac.jp/) websites, and DeepMind’s [AlphaCode paper](https://arxiv.org/pdf/2203.07814.pdf) refers to the CodeNet dataset as a “competitive programming dataset.” This dataset deliberately contains a wide variety of different submissions, including early drafts that coders improved over time, as well as submissions from both beginners and experts. This diversity makes the dataset useful as a performance optimization benchmark, since we can examine how to help beginners and experienced programmers improve the performance of their code.
>
> For reference, here is the URL to the problem in Figure 4: [B - One Clue](https://atcoder.jp/contests/abc137/tasks/abc137_b)
>
> Here are some other URLs to problems in the PIE test set:
>
> - [F - Fractal Shortest Path](https://atcoder.jp/contests/panasonic2020/tasks/panasonic2020_f)
>
> - [E - Jigsaw](https://atcoder.jp/contests/agc017/tasks/agc017_e)
>
> - [Problem B: Compress Files](https://judge.u-aizu.ac.jp/onlinejudge/description.jsp?id=2080)
>
> ---
>
> > ### **[R3-Q11] Fig 6b is missing an obvious break statement to help performance**
>
> Thanks for pointing out the additional opportunity for optimization. While the models are trained to generate faster programs, there may still be room for improvement. We will clarify this point in the caption that: `the example on the right is faster, but the generated code could have been even faster if it included a break statement.`
>
> ---
>
> > ### **[R3-Q12] "The evaluation setup is such that only good generations are considered in calculating the metric.**
>
> This strategy reflects the natural way a program optimization tool would be used: keep the best program collected, including the initial program. In particular, if a programmer realizes the generated program is incorrect or slower, they can simply revert to the original and have a speedup of 1.0. This approach aligns with standard software development practices, where unproductive changes are discarded. In fact, for GPT-3.5 with self-play (SP), this strategy actually has less impact than other models because so many programs are optimized: GPT-3.5 (SP) improved performance for 87.68% of programs in the benchmark by at least 10%. Thus, the strategy was only applied to a small fraction of programs (at most 12.32%).
>
> ---
> ---
>
> ## [Go to Reviews](https://openreview.net/forum?id=ix7rLVHXyY&noteId=vLeyg6r8nW)
>
> ---
> ---

---

> ### Author Response · Authors · 2023-11-22
>
> We hope that you are satisfied with the answers we have provided. As the discussion period comes to an end, we would be grateful if you could let us know if we have adequately addressed your comments and whether you have any further questions.

---

> ### Comment · Reviewer_VFMY · 2023-11-22
> **thanks for the explanations**
>
> I have read the rebuttal. Thanks for providing links to the original problems of some examples used, and they help clarify some of the doubts. I do still have doubts on the data quality as well as the claim of 6.86X speed up versus 4.06X of best human solution. Maybe showing examples of model-generated vs best-human-solution would help; maybe reporting percentiles in additional to averages will help. I am also not convinced by explanation of correctness percentage, which is again a data quality doubt.
> I have updated my rating based on the new information.

---

> ### Author Response · Authors · 2023-11-23
> **Thanks for your time and feedback!**
>
> Dear Reviewer VFMY,
>
> We are pleased that we could clarify doubts you had and that you found the information we provided useful. Thank you again for all your time, which will help us improve our work. With the time remaining, we’ve tried to provide the following responses below to your recent suggestions.
>
>
>
> > ### **[R3-Q13] Maybe showing examples of model-generated vs best-human-solution would help**
>
> The following are some examples where the model-generated solution was at least 5$\times$ faster than the best human solution:
>
> ---
> ---
>
> ### *Example 1 $\rightarrow$ CodeLlama 13B with Performance Conditioning*
>
> Fastest Human Solution ($\mathbf{\underline{5.2\times}}$) vs. Model Generated Solution ($\mathbf{\underline{26.9\times}}$)
>
> $\downarrow\downarrow$ Fastest Human Solution for the Submission (5.2$\times$) $\downarrow\downarrow$
>
> ```cpp
> #include <iostream>
> using namespace std;
>
> int main()
> {
>     int s;
>     cin >> s;
>     int list[1000000];
>     int index = 0;
>     int a_i;
>     int a_pre = s;
>     list[0] = s;
>     index = index + 1;
>     bool search_flag = false;
>     while( true){
>         if ( a_pre % 2 == 0){
>             a_i = a_pre / 2;
>         } else {
>             a_i = 3 * a_pre + 1;
>         }
>         list[index] = a_i;
>         index = index + 1;
>         for ( int i = 0; i< index-1; ++ i ){
>             if ( list[i] == a_i) {
>                 search_flag = true;
>                 break;
>             }
>         }
>         if (search_flag==true){
>             break;
>         }
>         a_pre = a_i;
>
>     }
>     cout << index << endl;
>     return 0;
> }
> ```
>
> $\downarrow\downarrow$ Model Generated Solution (26.9$\times$) $\downarrow\downarrow$
>
> ```cpp
> #include<stdlib.h>
>
> int main(){
>   int s;
>   scanf("%d", &s);
>   int* a = (int*)calloc(1000000, sizeof(int));
>   a[s]++;
>   for(int i=2; i<1000000; i++){
>     if(s%2 == 0){
>       s /= 2;
>       a[s]++;
>     }else{
>       s *= 3;
>       s++;
>       a[s]++;
>     }
>     if(a[s] == 2){
>       printf("%d", i);
>       break;
>     }
>   }
> }
> ```
> ---
> ---
>
> ### *Example 2 $\rightarrow$ GPT3.5 Fine-Tuned with Synthetic Data*
>
> Fastest Human Solution ($\mathbf{\underline{5.12\times}}$) vs. Model Generated Solution ($\mathbf{\underline{27.5\times}}$)
>
> $\downarrow\downarrow$ Fastest Human Solution for the Submission (5.12$\times$) $\downarrow\downarrow$
>
> ```cpp
> #include<bits/stdc++.h>
> using namespace std;
> #define int long long
> int editDistance(string x,string y){
>   int n=x.length(),m=y.length();
>   int INF=1LL<<55LL;
>   vector<vector<int> > dp(n+1,vector<int>(m+1,INF));
>   for(int i=0;i<=n;i++) dp[i][0]=i;
>   for(int i=0;i<=m;i++) dp[0][i]=i;
>   for(int i=1;i<=n;i++){
>     for(int j=1;j<=m;j++){
>       dp[i][j]=min(dp[i][j],dp[i-1][j]+1);
>       dp[i][j]=min(dp[i][j],dp[i][j-1]+1);
>       dp[i][j]=min(dp[i][j],dp[i-1][j-1]+(x[i-1]!=y[j-1]));
>     }
>   }
>   return dp[n][m];
> }
> signed main(){
>   string s1,s2;
>   cin>>s1>>s2;
>   cout<<editDistance(s1,s2)<<endl;
>   return 0;
> }
> ```
>
> $\downarrow\downarrow$ Model Generated Solution (27.5$\times$) $\downarrow\downarrow$
>
> We note that the problem description restricts inputs to be within a length of 1000.
>
> ```cpp
> #include <cstdio>
> #include <cstring>
> const int N=1005;
> int n,m,dp[N][N];
> char a[N],b[N];
> int min(int a,int b){return a<b?a:b;}
> int main()
> {
> 	scanf("%s%s",a+1,b+1);
> 	n=strlen(a+1),m=strlen(b+1);
> 	for(int i=1;i<=n;i++) dp[i][0]=i;
> 	for(int i=1;i<=m;i++) dp[0][i]=i;
> 	for(int i=1;i<=n;i++)
> 		for(int j=1;j<=m;j++)
> 		{
> 			dp[i][j]=dp[i-1][j-1];
> 			if(a[i]!=b[j]) dp[i][j]++;
> 			dp[i][j]=min(dp[i][j],min(dp[i-1][j]+1,dp[i][j-1]+1));
> 		}
> 	printf("%d",dp[n][m]);
> 	return 0;
> }
> ```
>
> ---
> ---
>
> ### *Example 3 $\rightarrow$ GPT3.5 Fine-Tuned with HQ-Only Data*
>
> Fastest Human Solution ($\mathbf{\underline{5.18\times}}$) vs. Model Generated Solution ($\mathbf{\underline{27.3\times}}$)
>
> $\downarrow\downarrow$ Fastest Human Solution for the Submission (5.18$\times$) $\downarrow\downarrow$
>
> ```cpp
> #include <iostream>
>
> int main()
> {
>     int s;
>     std::cin >> s;
>     int list[1000000];
>     int index = 0;
>     int a_i;
>     int a_pre = s;
>     list[0] = s;
>     index = index + 1;
>     bool search_flag = false;
>     while( true){
>         if ( a_pre % 2 == 0){
>             a_i = a_pre / 2;
>         } else {
>             a_i = 3 * a_pre + 1;
>         }
>         list[index] = a_i;
>         index = index + 1;
>         for ( int i = 0; i< index-1; ++ i ){
>             if ( list[i] == a_i) {
>                 search_flag = true;
>                 break;
>             }
>         }
>         if (search_flag==true){
>             break;
>         }
>         a_pre = a_i;
>
>     }
>     std::cout << index << std::endl;
>     return 0;
> }
> ```
>
> $\downarrow\downarrow$ Model Generated Solution (27.3$\times$) $\downarrow\downarrow$
>
> ```cpp
> #include <stdio.h>
>
> int f(int n) {
> 	if (n % 2 == 0) {
> 		return n / 2;
> 	} else {
> 		return 3 * n + 1;
> 	}
> }
>
> int main() {
>   int s, count = 1;
>   scanf("%d", &s);
>
>
>   while (true) {
>   	if (s == 4 || s == 2 || s == 1) {
>   		count += 3;
>   		break;
> 	  }
>   	s = f(s);
>   	count++;
>   }
>   printf("%d", count);
>   return 0;
> }
> ```

---

> ### Author Response · Authors · 2023-11-23
>
> > ### **[R3-Q14] maybe reporting percentiles in additional to averages will help**
>
> Thank you for your suggestion. The table below reports the fraction of all generations falling in 10 different speedup categories. The majority of generations are $>=$ 5.0$\times$ speedup and $<$ 10.0$\times$ speedup, which helps to explain the 6.86$\times$ result.
>
> | Speedup   | % Percent Programs with the Speedup (Best@8) |
> |---------------------|--------------------------------------|
> | >= 1.1$\times$              | 87.68%                               |
> | >= 1.25$\times$             | 87.07%                               |
> | >= 1.5$\times$              | 82.89%                               |
> | >= 1.75$\times$             | 82.59%                               |
> | >= 2.0$\times$              | 79.12%                               |
> | >= 2.5$\times$              | 77.39%                               |
> | >= 3.0$\times$              | 76.07%                               |
> | >= 4.0$\times$              | 74.64%                               |
> | >= 5.0$\times$              | 73.01%                               |
> | >= 10.0$\times$             | 9.06%                                |
>
>
> We hope that these responses provide further clarification. Thanks again for your feedback.
>
>
> Best,
>
> Authors

---

### Official Review · Reviewer_nenm · 2023-11-01

**Soundness:** 3 good
**Presentation:** 3 good
**Contribution:** 3 good
**Rating:** 8
**Confidence:** 3

**Summary:**

This paper proposes a new benchmark for training and evaluating LLMs to generate performance-improving code edits given an existing unoptimized program as well as an evaluation of current and novel prompting/finetuning methods for adapting LLMs for this task. The benchmark is constructed from CodeNet tasks and the authors annotate the runtimes using the gem5 environment along with caveats that this performance measurement can be very difficult in other benchmarking setups. The authors show improvements over the human baselines on speedups with a variety of methods, and show a systematic ablation with various models, prompting, and retrieval schemes on both open-source and private large language models.

**Strengths:**

The benchmark is an important artifact that the community will continue to build upon, especially as code-generating/editing large language models continue to be developed and deployed in research and production environments. The analysis and ablations are very thorough and further justify the benchmark and prompting strategies as important contributions.

**Weaknesses:**

The experiments are very thorough, but it seems that the correctness of the models degrades with introduced methods. While it seems that the paper's primary contribution is the dataset, further analysis of correctness (as opposed to just pure speedups/optimization %) would further solidify the adaptation methods sections of the paper.

**Questions:**

Are the speedups reported only on edited programs that are verified to be correct?
How does the model generalization scale with the amount of synthetic data used?

---

> ### Author Response · Authors · 2023-11-17
>
> Thank you for taking the time to review the paper and provide feedback! We are happy that you believe the benchmark is an important artifact that will be useful to the broader community. We're also glad that you think the methods evaluated are comprehensive.
>
> ---
>
> > ### **[R2-Q1] Are the speedups reported only on edited programs that are verified to be correct?**
>
> Yes, only speedups from programs that pass all test cases are included. If a program is incorrect even for a single unit test, we discard it and do not include it in calculating speedup.
>
> ---
>
> > ### **[R2-Q2] How does the model generalization scale with the amount of synthetic data used?**
>
> We are running another experiment to investigate this, and we will report the result. We will fine-tune with 50% less synthetic data to explore this.
>
> ---
>
> > ### **[R2-Q3] Where do the LLMs fail and why?**
>
> Thank you for the suggestion to investigate error analysis! We performed this analysis for GPT-3.5 fine-tuned with synthetic data. Specifically, we analyzed the generated programs that PIE fails to optimize and the root cause of each failure. Our findings are summarized below:
>
> | Result | Percentage |
> |-|-|
> | Failed to compile (syntax/type errors) | 12.51% |
> | Compiled, but got [95-100%] of test cases wrong | 27.59% |
> | Compiled, but got (75, 95%] of test cases wrong | 12.08% |
> | Compiled, but got (25, 75%] of test cases wrong | 10.84% |
> | Compiled, but got (0-25%] of test cases wrong (at least 1 test case was wrong) | 6.90% |
> | Ran all test cases, but the program was slower than the original | 9.93% |
> | Ran all test cases, but the program was the same speed as the original | 9.40% |
> | Ran all test cases, the program was faster, but not 1.1$\times$ speedup or higher | 10.75% |
>
> In summary, a large fraction (~60%) of the failures happen because the proposed changes break a unit test. In about 30% of the cases, the model produces a correct, but the generated program is either slower (10%) or doesn't meet our threshold for speedup (10%). Finally, in about 10% of the cases, the generated program has a syntax error.
> Additionally, with test cases, we find that when the model gets the program wrong, it seems to most often get it "quite wrong" by missing most test cases. We will update the existing analysis (Appendix A) with these new findings.
>
>
> Finally, we conduct additional analysis to investigate the properties of programs that PIE fails to optimize. The results show a mild negative correlation between the problem description length and average accuracy (-0.15) and between the source program length and average accuracy (-0.26), suggesting longer inputs slightly reduce accuracy. Additionally, the average speedup has a mild negative correlation with both the problem description length (-0.16) and the source program length (-0.11), indicating a minimal impact of length on speedup compared to correctness. Overall, this analysis reveals that language models struggle to generate a correct program when faced with larger source programs and challenging problems, but their ability to optimize programs is minimally impacted. This motivates a future work direction where techniques from program repair may be combined with PIE for better results.
>
>
> ---
> ---
>
> ## [Go to Reviews](https://openreview.net/forum?id=ix7rLVHXyY&noteId=ieRhpxAzdk)
>
> ---
> ---

---

> ### Author Response · Authors · 2023-11-21
> **Results for [R2-Q2] $\rightarrow$ Using less synthetic data**
>
> > ### **[R2-Q2-Results] How does the model generalization scale with the amount of synthetic data used?**
>
> We’ve obtained the results from this experiment: we trained the same GPT3.5 Model with 50% of the synthetic data pairs and included it in the tables below (Experiment: HQ + 50% Self-Play).
>
>
> **Best@1 Results**
>
> | Experiment   | Setup      | %Opt (>1.10 speedup) | Average Speedup | %Correct |
> |--------------------|------------|----------------------|-----------------|----------|
> | HQ-Only            | Fine-tuned | 38.49%               | 2.70            | 59.16%   |
> | HQ + 50% Self-Play | Fine-tuned | 40.22%               | 2.87            | 58.35%   |
> | HQ + Self-Play     | Fine-tuned | 45.62%               | 3.02            | 61.71%   |
>
>
> **Best@8 Results**
>
> | Experiment   | Setup      | %Opt (>1.10 speedup) | Average Speedup | %Correct |
> |--------------------|------------|----------------------|-----------------|----------|
> | HQ-Only            | Fine-tuned | 86.66%               | 6.74            | 95.42%   |
> | HQ + 50% Self-Play | Fine-tuned | 85.95%               | 6.47            | 94.70%   |
> | HQ + Self-Play     | Fine-tuned | 87.68%               | 6.86            | 95.11%   |
>
>
> Best@1, the results for %Opt, Speedup, and %Correct interpolate between using all of the synthetic data and none of the synthetic data. Best@8; the model fine-tuned with less synthetic data seems to underperform both models.
>
> Although we use the same hyperparameters for sampling, HQ + 50% Self-Play seems to generate with less diversity  and scales poorly with more generations, potentially due to a lack of diversity in the new subset of training data.
>
> ---
> ---
>
> ## [Go to Reviews](https://openreview.net/forum?id=ix7rLVHXyY&noteId=ieRhpxAzdk)
>
> ---
> ---

---

> ### Author Response · Authors · 2023-11-22
>
> We hope that you are satisfied with our answers and the additional results we have provided. As the discussion period comes to an end, we would be grateful if you could let us know if we have adequately addressed your comments and whether you have any further questions.

---

> > ### Comment · Reviewer_nenm · 2023-11-22
> >
> > Thank you for the thorough follow-up and analysis! Thank you for clarifying when and why the failures happen in your evaluation. The follow-up experiment on scaling down the amount of synthetic data used is very interesting, although it would be more informative if we also saw the result of increasing the amount of self-play data (e.g. 200%), does it continue to improve?
> >
> > Thank you for answering these questions. I will maintain my score.

---

> ### Author Response · Authors · 2023-11-23
> **Thanks for your time and feedback!**
>
> Dear Reviewer nenm,
>
>
> Thanks for your time and feedback, which will help us improve our work. We are pleased to see that we were able to address your concerns. Thank you for the suggestion on increasing the amount of self-play data to 200%. As the experiment will take around 4 days and some extra budget, we will add these results in the next version.
>
> Best,
> Authors

---

### Official Review · Reviewer_8KXu · 2023-11-04

**Soundness:** 3 good
**Presentation:** 2 fair
**Contribution:** 3 good
**Rating:** 8
**Confidence:** 3

**Summary:**

The paper proposes PIE dataset. Each sample in the dataset is basically a pair, showing a program and its optimized counterpart.
The impact of PIE dataset is further studied on several LLMs (both open and closed), including the family of CodeLlama and GPT3.5 and GPT-4. Moreover, a broad range of adaptations is used to improve the results of the LLMs, such as instruction, few shot, chain of thoughts, dynamic few shot, and fine-tuning.
The experimental results indicate that each adaptation strategy improves the results, with fine-tuning having the most significant improvement over the others.

**Strengths:**

+ The problem of performance editing is important. Not only should the generated code be correct, but also it should be as efficient as possible. Therefore, the paper targets an important problem.

+ The range of adaptations considered is wide, and the experimental results give the reader a clear image of how each adaptation strategy can be used to improve the results.

+ The paper uses top closed-sourced (GPT3.5 and 4) and open-sourced (CodeLLama 7B, 13B, 34B) to study the impact of PIE dataset. It clearly shows the gap between closed and open LLMs and how different adaptations can narrow this gap.

**Weaknesses:**

- The study mostly focuses on the percentage of performance, and in cases where the runtime of the generated program is slower or the code is incorrect, speed up is considered one. This is not a good approach. In particular, a study is needed on what category of programs LLMs often fail to produce correct code or optimized code. In such a study, it will be easier to infer in which cases it is better not to use the LLMs. Or which category of programs needs to be further improved.

- For synthetic data, it is mentioned that semantic duplicates are tracked, but no information is provided on how semantic similarity is conducted. Is it also based on CodeBertScore?

- It is interesting to see that Perf-Cond has decreased the percentage of correctness. Any reason why Perf-Cond degrades the ability of the LLMs to produce the correct code?

-  The presentation could be improved as an example, Table 1 is not referenced anywhere in the text, or in Table 2, it is better to group the Dynamic Retrievals together. The additional horizontal line is confusing.

- In Table 2, why is the choice of models not consistent across different methods? Like, there is no few shots learning or COT for GPT-4.

**Questions:**

For detailed questions, please refer to the weakness section.

- The authors have not discussed enough about the cases where the LLMs fail or why they fail. Could they add further experiments and clarify it?

- In Table 2, why is the choice of models inconsistent across different methods? Like, there are no few shot learning or COT for GPT-4.

- Why does Perf-Cond degrade the ability of the LLMs to produce the correct code?

- How semantic similarity is conducted where semantic duplicates are tracked?

---

> ### Author Response · Authors · 2023-11-17
>
> Thank you for taking the time to review our paper. We were happy to read that you find the problem area important and that we experiment with a large range of models and techniques. We think that all your questions are addressable within this discussion period. Please see our response below. We would love to address additional questions during the discussion period if anything needs to be clarified.
>
> ---
>
> > ### **[R1-Q1] Where do the LLMs fail and why?**
>
> Thank you for the suggestion to investigate error analysis!
> We performed this analysis for GPT-3.5 fine-tuned with synthetic data. Specifically, we analyzed the generated programs that PIE fails to optimize and the root cause of each failure. Our findings are summarized below:
> | Result | Percentage |
> |-|-|
> | Failed to compile (syntax/type errors) | 12.51% |
> | Compiled, but got [95-100%] of test cases wrong | 27.59% |
> | Compiled, but got (75, 95%] of test cases wrong | 12.08% |
> | Compiled, but got (25, 75%] of test cases wrong | 10.84% |
> | Compiled, but got (0-25%] of test cases wrong (at least 1 test case was wrong) | 6.90% |
> | Ran all test cases, but the program was slower than the original | 9.93% |
> | Ran all test cases, but the program was the same speed as the original | 9.40% |
> | Ran all test cases, the program was faster, but not 1.1$\times$ speedup or higher | 10.75% |
>
>
> In summary, a large fraction (~60%) of the failures happen because the proposed changes break a unit test. In about 30% of the cases, the model produces a correct, but the generated program is either slower (10%) or doesn't meet our threshold for speedup (10%). Finally, in about 10% of the cases, the generated program has a syntax error. Additionally, with test cases, we find that when the model gets the program wrong, it seems to most often get it "quite wrong" by missing most test cases. We will update the existing analysis (Appendix A) with these new findings.
>
> Finally, we conduct additional analysis to investigate the properties of programs that PIE fails to optimize. The results show a mild negative correlation between the problem description length and average accuracy (-0.15) and between the source program length and average accuracy (-0.26), suggesting longer inputs slightly reduce accuracy. Additionally, the average speedup has a mild negative correlation with both the problem description length (-0.16) and the source program length (-0.11), indicating a minimal impact of length on speedup compared to correctness. Overall, this analysis reveals that language models struggle to generate a correct program when faced with larger source programs and challenging problems, but their ability to optimize programs is minimally impacted. This motivates a future work direction where techniques from program repair may be combined with PIE for better results.
>
> ---
>
> > ### **[R1-Q2] Why did we leave out Few-shot and CoT for GPT4?**
>
> At the time of writing, we did not have the funds to run all experiments with GPT-4. We were able to get the funds, and we have evaluated the outputs, and we will include them in the draft. For GPT4, both Few Shot and CoT prompting improve over Instruction prompting but underperform retrieval-based prompting. For example, Best@4 for CoT has a %Opt of 38.60% and a speedup of 1.51$\times$, while Few-Shot has a %Opt of 30.83% and a speedup of 1.37$\times$.
> | Prompting Method | %Opt | Speedup |
> |------------------|--------|---------|
> | Instruction Only (Best@4) | 16.58% | 1.23$\times$ |
> | CoT (Best@4) | 38.60% | 1.51$\times$ |
> | Few-Shot (Best@4) | 30.83% | 1.37$\times$ |
> | Dynamic Retrieval (Best@4) | 69.03% | 3.56$\times$ |
>
> ---
>
> > ### **[R1-Q3] Why does Perf-Cond degrade the LLM’s ability to produce correct code?**
>
> We believe that conditioning the model only to generate programs with a 10/10 optimization rate may constrain the number of optimizations available for any given input. To verify this, we experimented using the first 6 generations from the 7b Perf-Cond model when conditioned on 10/10 versus combining the first 2 generations when conditioned on 10/10, 9/10, and 8/10. When we did this, we saw a %Correct increase from 59.95% to 64.36%. These figures support the explanation that performance labels may restrict which generated programs are correct.
>
> ---
> ---
>
> ## [Go to Reviews](https://openreview.net/forum?id=ix7rLVHXyY&noteId=rgBN5tF6tF)
>
> ---
> ---

---

> ### Author Response · Authors · 2023-11-17
>
> > ### **[R1-Q4] How is filtering out semantic duplicates conducted for Self-Play?**
>
> In Section A.4 of the Appendix, we describe how we check semantic similarity. In detail, because our goal is to only demonstrate inequivalence, not equivalence, we use a set of test cases, and ensure that at least 1 output is different from programs in the PIE dataset. We use 200 test cases for all programs, and the generated program must run without errors on at least 20 of these. To achieve this, we had to execute more than 1.4 million binary input pairs and track all the outputs.
>
> ---
>
> > ### **[R1-Q5] The presentation could be improved as an example...**
>
> Thank you for pointing these presentation details out to us; we will update the draft based on your feedback!
>
> ---
>
> > ### **[R1-Q6] “1.00” is used for speedup when a generation is incorrect or slower**
>
> This strategy reflects the natural way a program optimization tool would be used: keep the best program collected, including the initial program. In particular, if a programmer realizes the generated program is incorrect or slower, they can simply revert to the original and have a speedup of 1.0. This approach aligns with standard software development practices, where unproductive changes are discarded.
>
> ---
> ---
>
> ## [Go to Reviews](https://openreview.net/forum?id=ix7rLVHXyY&noteId=rgBN5tF6tF)
>
> ---
> ---

---

> ### Author Response · Authors · 2023-11-22
>
> We hope that you are satisfied with our answers and the additional results we have provided. As the discussion period comes to an end, we would be grateful if you could let us know if we have adequately addressed your comments and whether you have any further questions.

---

> > ### Comment · Reviewer_8KXu · 2023-11-22
> >
> > I read the rebuttal. My major concerns are clarified, and thank you for the explanation.

---

> ### Author Response · Authors · 2023-11-23
> **Thanks for your time and feedback!**
>
> Dear Reviewer 8KXu,
>
> Thanks for your time and feedback, which will help us improve our work. We are pleased to see that we were able to address your concerns.
>
> Best,
>
> Authors

---

### Author Response · Authors · 2023-11-21
**Thanks for the feedback!**

We would like to thank all the reviewers for their valuable feedback. We are encouraged that the reviewers found our work to be well motivated (Reviewers [VFMY](https://openreview.net/forum?id=ix7rLVHXyY&noteId=vLeyg6r8nW), [jit6](https://openreview.net/forum?id=ix7rLVHXyY&noteId=zNxhW6pwmc)), recognized the problem's importance and the dataset's impact to be significant (Reviewers [8KXu](https://openreview.net/forum?id=ix7rLVHXyY&noteId=rgBN5tF6tF), [nenm](https://openreview.net/forum?id=ix7rLVHXyY&noteId=ieRhpxAzdk), [jit6](https://openreview.net/forum?id=ix7rLVHXyY&noteId=zNxhW6pwmc)), and considered our experiments to be comprehensive (Reviewers [nenm](https://openreview.net/forum?id=ix7rLVHXyY&noteId=ieRhpxAzdk), [jit6](https://openreview.net/forum?id=ix7rLVHXyY&noteId=zNxhW6pwmc)). We have addressed reviewers' comments individually and look forward to comments within the remainder of the author response period.

We have already obtained GPT-4 evaluation results that reviewers [jit6](https://openreview.net/forum?id=ix7rLVHXyY&noteId=zNxhW6pwmc) and [8KXu](https://openreview.net/forum?id=ix7rLVHXyY&noteId=rgBN5tF6tF) have requested (`Page-7-Table-2` and `Page-8-Table-3`)  and also shared the error analysis suggested by reviewers [8KXu](https://openreview.net/forum?id=ix7rLVHXyY&noteId=rgBN5tF6tF) and [nenm](https://openreview.net/forum?id=ix7rLVHXyY&noteId=ieRhpxAzdk) (`Page-18-Appendix-A.4-Table-5`). We have also updated the draft with the GPT-4 results, error analysis, as well as clarifications and style changes pointed out during the discussion period (`Page-3-Section-2`, `Page-17-Appendix-A.3`, `Page-22-Appendix-A.10`). We also evaluated and reported GPT-4 results with 8 generations, like all other models, previously our budget constrained us to only 4 generations (`Page-8-Table-3`). These changes are in $\underline{\color{magenta}{magenta}}$ throughout the updated draft. We welcome any further comments and points for clarification from the reviewers.

---

### Meta-Review · Area_Chair_fuZF · 2023-12-09

**Metareview:**

This paper proposes a new benchmark, Performance Improving Edits (PIE),  for training and evaluating LLMs to generate performance-improving code edits. The capabilities of various LLMs for this task are evaluated using promoting and fine-tuning methods.

I thank the authors and the reviewers for the engaging discussions. The reviewers agree that the paper targets an important problem, presenting a wide range of adaptations. One of the reviewers emphasized that the benchmarks is an important artifact that the community will continue to build upon. In their rebuttal, the authors provided an insightful error analysis. The only remaining issues stem from one of the reviewers questioning the quality of the generated dataset and the speed-up claims. I believe the authors have addressed these satisfactorily. Hence, I recommend acceptance.

**Justification For Why Not Higher Score:**

Potentially interesting to a wide part of the ML community but there may be still questions about the actual quality of the generated dataset that make it harder to predict the impact of this paper.

**Justification For Why Not Lower Score:**

Interesting problem addressed with modern and popular LLM techniques.

---

### Decision · Program_Chairs · 2024-01-16

Accept (spotlight)